# PuzzleMoE: Efficient Compression of Large Mixture-of-Experts Models via Fine-Grained Expert Merging and Bit-packed Inference

**Yushu Zhao** [* † 1]  **Zheng Wang** [* 2]  **Minjia Zhang** [2]

## Abstract

Mixture-of-Experts (MoE) have shown strong potential in scaling language models efficiently by activating only a small subset of experts per input. However, their deployment remains limited due to the high memory overhead associated with storing all expert parameters, particularly as the number of experts increases. To address this challenge, prior works have explored expert dropping and merging strategies; however, they often suffer from notable performance drop especially at high compression ratios due to their reliance on coarse-grained tensor- or expert-level operations. In this paper, we introduce PuzzleMoE, the first MoE merging method to enable fine-grained element-wise merging while achieving both high accuracy and inference speed, via two key innovations: First, PuzzleMoE performs sparse expert merging by identifying element-wise weight redundancy and specialization. It introduces a dual-mask approach to capture both shared and expert-specific salient parameters. Second, to avoid the overhead of storing masks and signs, we introduce a bit-packed encoding scheme that reuses underutilized exponent bits, enabling efficient MoE inference on GPUs. Extensive experiments demonstrate that PuzzleMoE outperforms prior MoE compression methods by up to 16.7% on MMLU at 50% compression ratio, and achieves up to 1.80× end-to-end inference throughput gain.

## 1. Introduction

Mixture-of-Experts (MoE) architectures have demonstrated remarkable scalability in language models by activating only a subset of expert sub-networks per input. However, deploying these models in real-world applications remains challenging due to their large memory footprint: all expert weights must be stored in memory, regardless of which subset is activated during inference. As the number of experts grows, this memory overhead becomes increasingly prohibitive, as evidenced by advances such as Mixtral (Jiang et al., 2024), DeepSeek-MoE (Dai et al., 2024), and Qwen-MoE (Team, 2024), especially for resource-constrained deployments. For instance, Mixtral-8x7B has 47 billion parameters, about 45 billion of which are in the expert modules, requiring at least two A100-80GB GPUs to load in 16-bit.

To reduce this memory cost, previous studies have investigated expert compression methods, including expert dropping and expert merging. Expert dropping methods (Lu et al., 2024; Lee et al., 2025b) remove entire experts considered less important based on their output over a calibration dataset. However, it is easy for them to accidentally discard important knowledge, leading to a sharp accuracy drop (Chen et al., 2025). In contrast, expert merging methods attempt to combine similar experts rather than removing them entirely, typically via expert clustering (Chen et al., 2025) or using low-rank approximations (Li et al., 2025). Although these methods generally outperform expert dropping, they still experience significant accuracy degradation, with performance drops of over 20% on MMLU (Chen et al., 2025; Li et al., 2025) as demonstrated in Figure 1.

The significant performance degradation indicates that existing **coarse-grained** expert dropping and merging strategies are insufficient. By operating at the level of entire experts or weight tensors, these methods fail to distinguish between shared parameters and expert-specific weights, leading to over-merging that hurts expert specialization and model quality. Moving beyond coarse-grained operations, fine-grained merging decisions have the potential to better preserve accuracy, but they introduce substantial challenges. First, making fine-grained merging decision requires navigating a large combinatorial space, making naive merging computationally intractable. Second, fine-grained sparsity often relies on explicit storage of masks or indices, which introduces additional metadata overhead that can significantly reduce or even negate the intended memory savings. In

---

[*]Equal contribution [†]Work done while intern at UIUC. [1]Tsinghua University [2]University of Illinois Urbana-Champaign. Correspondence to: Zheng Wang <zhengw10@uiuc.edu>, Minjia Zhang <minjiaz@uiuc.edu>.

*Proceedings of the 43rd International Conference on Machine Learning*, Seoul, South Korea. PMLR 306, 2026. Copyright 2026 by the author(s).

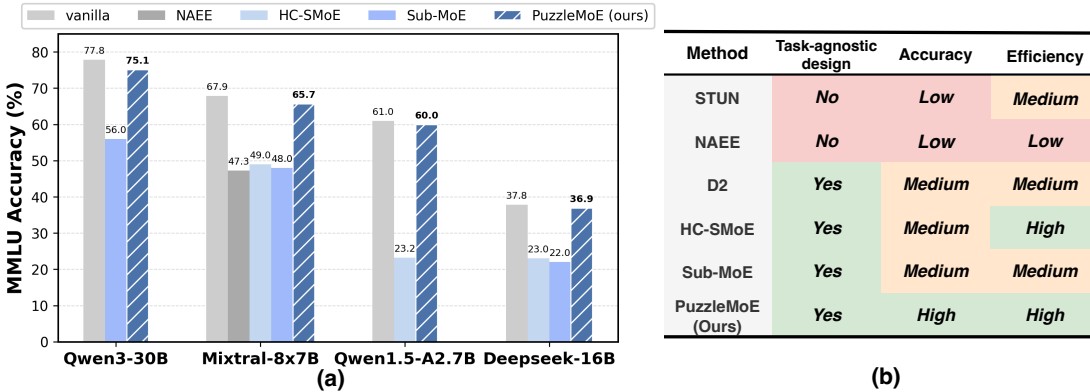

*Figure 1.* (a): Accuracy of different MoE models on MMLU benchmark under 50% compression ratio with various expert compression methods, among which PuzzleMoE achieves the best accuracy. (b): Comparison of different expert compression methods, among which PuzzleMoE effectively and efficiently retains MoE model performance after compression.

addition, many existing expert compression methods rely on task-specific calibration (Lu et al., 2024; Lee et al., 2025b) and require extensive offline compression time, limiting their practicality and scalability (Section 4.4).

In this paper, we propose PuzzleMoE, the first MoE merging method to enable element-wise merging while achieving both high accuracy and inference speed. First, we introduce a novel fine-grained expert merging algorithm that selectively merges experts at element-wise level. Specifically, we construct two masks: (1) an **element-wise similarity mask** that identifies weights with high magnitude similarity between expert pairs; and (2) an **activation-weight saliency mask** that identifies weights critical to each expert's unique specialization. This dual-mask design allows PuzzleMoE to merge only redundant parameters while preserving expert-specific capacity by retaining salient weights.

While element-wise merging better preserves both shared and expert-specific weights, storing element-wise binary masks and sign information for merged expert can introduce substantial metadata overhead, which negates the memory benefits of fine-grained compression at scale. To overcome this challenge, we introduce a bit-packed encoding scheme that reuses underutilized bits in floating-point representations. We observe that the exponent field of expert weights typically occupies a narrow range during inference, leaving multiple bits underutilized. PuzzleMoE leverages these bits to embed binary masks and sign bits directly into the weight tensors, eliminating the need for auxiliary metadata storage. To support this encoding during inference, we further design a custom decode-GEMM kernel that performs on-the-fly decoding with the matrix multiplication operation, enabling fast and memory-efficient execution of PuzzleMoE. Our contributions are summarized as follows:

- We propose PuzzleMoE, the first MoE merging method to enable fine-grained, element-wise expert merging

that achieves both high accuracy and fast inference speed. Unlike prior expert merging approaches operating at coarse granularity, PuzzleMoE selectively merges redundant parameters and preserves expert-specific weights through a dual-mask design based on weight similarity and activation-weight saliency.

- We design a novel bit-efficient encoding scheme that embeds masks and signs directly into floating-point weights, eliminating auxiliary metadata storage and enabling efficient MoE inference via a lightweight custom decode-GEMM kernel.

- We evaluate PuzzleMoE across four MoE models and nine benchmarks, demonstrating consistent improvements over prior methods. In particular, PuzzleMoE achieves up to 16.7% higher accuracy than prior methods under 50% compression ratio, and up to 1.80× higher end-to-end throughput on Mixtral-8x7B.

## 2. Related Works

Quantization and pruning are two widely adopted orthogonal techniques for model compression and have recently attracted increasing attention in the context of MoE LLMs (Chen et al., 2025; Gu et al., 2025; Huang et al., 2025; Duanmu et al., 2025; Hu et al., 2025; Li et al., 2025). Quantization methods (Frantar et al., 2023; Lin et al., 2024) exploit per-weight redundancy to reduce memory footprint by lowering weight precision, achieving substantial compression with minimal accuracy degradation. In contrast, pruning-based approaches including expert dropping and merging remain less mature. Existing methods often suffer from severe performance degradation at moderate sparsity levels; for instance, pruning 50% of experts can result in over 18.7% accuracy drop on the MMLU benchmark for Mixtral-8×7B (Chen et al., 2025; Gu et al., 2025; Li et al., 2025;

Lu et al., 2024). These observations indicate that effective pruning of MoE models remains an open challenge.

**Expert Dropping.** Expert dropping methods reduce MoE model size by removing entire expert modules considered unimportant. For example, NAEE (Lu et al., 2024) performs an exhaustive search to identify a subset of experts to retain, while STUN (Lee et al., 2025b) accelerates this process by exploiting latent behavioral similarity among experts, reducing the selection complexity to $O(1)$. Despite their effectiveness, these methods rely heavily on task-specific calibration datasets, as different downstream tasks often require different subsets of experts to be preserved. This limits their generality across tasks. Relatedly, MoE-I² (Yang et al., 2024) adopts a two-stage strategy combining inter-expert pruning with intra-expert low-rank decomposition. However, it requires finetuning to recover performance.

**Expert Merging.** To better preserve accuracy, several recent works have proposed merging similar experts rather than dropping them. Methods like HC-SMoE (Chen et al., 2025) use hierarchical clustering based on expert output similarity to identify and combine experts, while MC-SMoE (Li et al., 2024), D2 (Gu et al., 2025), and Sub-MoE (Li et al., 2025) adopt multi-stage merging pipelines, e.g., first merging experts based on similarities, and then adding low-rank matrices to approximate the residual information. While these methods generally outperform expert dropping in accuracy, they often require complex procedures like SVD. In addition, they rely on coarse-grained expert merging, which risks hurting the distinctions between specialized experts and suffers from severe accuracy drop at large compression ratios. In contrast, PuzzleMoE performs fine-grained, element-wise merging in a training-free and single-pass manner, effectively preserving expert diversity.

## 3. Method

In this section, we first elaborate on the motivation for adopting fine-grained, element-wise expert merging. We then introduce PuzzleMoE, which enables efficient fine-grained element-wise expert merging through a carefully designed pairwise dual-mask merging algorithm and a system-level co-design with bit-level packing.

### 3.1. Motivation: Element-wise Similarity in Experts' Weights

Prior studies (Chen et al., 2025; Gu et al., 2025; Li et al., 2025; Lu et al., 2024) have primarily focused on coarse-grained expert pruning or merging strategies. Such approaches inherently overlook potential similarities at a finer granularity within expert parameters. Motivated by this gap, we conduct a fine-grained analysis of weight-level similarity across experts in MoE LLMs and empirically observe the

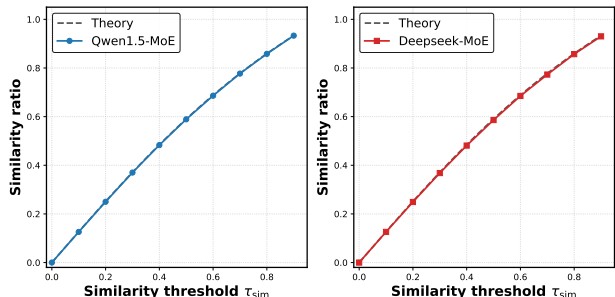

*Figure 2.* Relation between similarity ratio and similarity threshold $\tau_{\text{sim}}$.

presence of substantial element-wise redundancies.

Following the empirical observations in (Kim et al., 2024; Dettmers et al., 2023), we approximate the weights of different experts using zero-mean Gaussian distributions, which is validated in Appendix A.5. These distributions facilitate the characterization of element-wise similarity within a probabilistic context. Based on this, we model a pair of corresponding weight elements $w_i$ and $w_j$ from expert weights $\mathbf{W}_i$ and $\mathbf{W}_j$ as independent samples drawn from similar underlying distributions. This abstraction enables a tractable quantitative characterization of element-wise similarity under a probabilistic model.

**Lemma 3.1** (Informal). *Consider two independently sampled expert weight elements $w_i$ and $w_j$, where each element is drawn from a zero-mean Gaussian distribution, $w_i \sim \mathcal{N}(0, \sigma_i^2)$ and $w_j \sim \mathcal{N}(0, \sigma_j^2)$. For a similarity threshold $\tau_{\text{sim}} \in (0, 1)$, the probability that two corresponding weight elements are similar in a relative sense satisfies*

$$P\left( \frac{\left| |w_i| - |w_j| \right|}{|w_i| + |w_j|} < \tau_{\text{sim}} \right) = \frac{2}{\pi} \arctan\left( \frac{1 + \tau_{\text{sim}}}{1 - \tau_{\text{sim}}} \right)$$
$$- \frac{2}{\pi} \arctan\left( \frac{1 - \tau_{\text{sim}}}{1 + \tau_{\text{sim}}} \right).$$

The formal derivation is provided in Appendix A.6. We further evaluate the alignment between this theoretical prediction and empirical statistics from real-world MoE LLMs. As illustrated in Figure 2, empirical similarity ratios from both Qwen1.5-MoE and DeepSeek-MoE closely align with the theoretical prediction over a wide range of similarity thresholds. This strong agreement between theoretical and empirical curves suggests that element-wise similarity is an intrinsic statistical property of expert weights.

### 3.2. Dual-Mask Fine-Grained Expert Merging

Our goal is to merge experts in a fine-grained manner that preserves both shared parameters and expert-specific capacity. To this end, we introduce a dual-mask formulation that decomposes expert parameters into two complementary

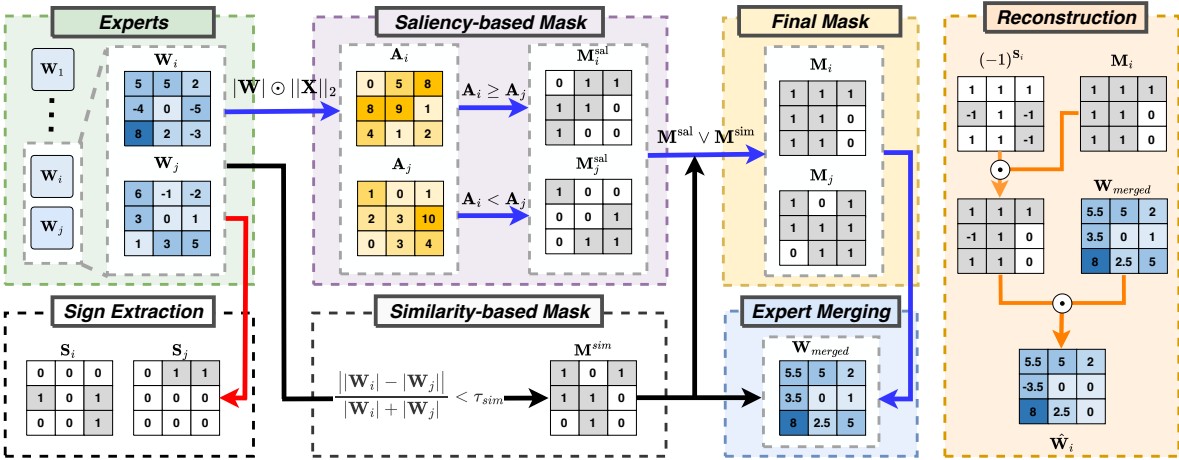

*Figure 3.* Overview of the fine-grained sparse expert merging algorithm. Given a pair of experts with weight matrices $\mathbf{W}i$ and $\mathbf{W}j$, we extract their sign matrices and compute element-wise saliency scores. A saliency-based mask $\mathbf{M}^{\text{sal}}$ selects dominant elements between experts, while a similarity-based mask $\mathbf{M}^{\text{sim}}$ identifies elements with high relative similarity under a threshold $\tau_{\text{sim}}$. The final merging mask $\mathbf{M}$ is obtained by combining these two masks, which is then used to produce the merged expert $\mathbf{W}_{\text{merged}}$.

components: shared parameters that can be safely merged, and salient parameters that should be preserved to maintain expert specialization.

Given two expert weight matrices $\mathbf{W}_i$ and $\mathbf{W}_j \in \mathcal{R}^{d \times h}$, which correspond to experts $\mathbf{E}_i$ and $\mathbf{E}_j$ from an MoE layer of a model with $N$ experts $\varepsilon = \{\mathbf{E}_1, \mathbf{E}_2, \ldots, \mathbf{E}_N\}$, we construct two element-wise masks. The first is an element-wise *similarity-based mask* that identifies shared weight elements. The second is a *saliency-based mask* used to preserve divergent yet important weight elements. Additionally, we store the original signs of each expert's weights and reapply them to $\mathbf{W}_{\text{merged}}$ during inference. The overall algorithm is illustrated in Figure 3. This dual-mask design enables fine-grained expert merging that explicitly distinguishes between shared and specialized parameters, avoiding the over-merging behavior of coarse-grained approaches.

**Similarity-based Mask.** Given two experts weights $\mathbf{W}_i$ and $\mathbf{W}_j$, we measure their per-element magnitude similarity using a symmetric percent difference (Miller, 1999):

$$\boldsymbol{\Delta} := \frac{||\mathbf{W}_i| - |\mathbf{W}_j||}{|\mathbf{W}_i| + |\mathbf{W}_j|},$$

where smaller values indicate greater similarity between two weight elements. It cleanly identifies elements in two experts' weights that are similar in magnitude regardless of direction to avoid spurious penalties from opposite signs. After that, the similarity-based mask can be defined as:

$$\mathbf{M}^{\text{sim}} := \mathbf{1}_{\{\boldsymbol{\Delta} \leq \tau_{\text{sim}}\}} \in \{0,1\}^{d \times h},$$

where $\tau_{\text{sim}} \in [0,1]$ is the pre-defined similarity threshold. $\mathbf{M}^{\text{sim}}$ is used to identify experts' elements with comparable magnitude so that they can be safely aggregated. We also

need to store each expert's sign pattern and reapply it to $\mathbf{W}_{merged}$ during inference, ensuring that shared merged elements can be reconstructed with minimal distortion:

$$\mathbf{S}_i := \mathbf{1}_{\{\mathbf{W}_i < 0\}} \in \{0,1\}^{d \times h}, \mathbf{S}_j := \mathbf{1}_{\{\mathbf{W}_j < 0\}} \in \{0,1\}^{d \times h}. \quad (1)$$

**Saliency-based Mask.** To decide which expert's elements to preserve, we extend the idea of quantifying the importance of weights by combining the magnitude of weights with the saliency of input activations in dense models (Sun et al., 2024) to MoE experts:

$$\mathbf{A}_i = |\mathbf{W}_i| \odot ||\mathbf{X}_i||_2, \qquad \mathbf{A}_j = |\mathbf{W}_j| \odot ||\mathbf{X}_j||_2, \quad (2)$$

where $\mathbf{X}$ represents a sample of input activations to a certain expert. Therefore, the saliency masks can be obtained as:

$$\mathbf{M}_i^{\text{sal}} := \mathbf{1}_{\{\mathbf{A}_i \geq \mathbf{A}_j\}} \in \{0,1\}^{d \times h}, \qquad \mathbf{M}_j^{\text{sal}} := \mathbf{1} - \mathbf{M}_i^{\text{sal}}. \quad (3)$$

These two masks indicate which expert has more important weights to be preserved at each element position.

**Expert Merging.** Given the dual-mask formulation, two experts can be merged as follows. Specifically, elements identified as similar by the similarity mask $\mathbf{M}_{\text{sim}}$ are averaged in magnitude, while dissimilar elements are selected from the more salient expert using the saliency masks. Formally, this sparse merging process can be expressed as:

$$\mathbf{M}_i = \mathbf{M}_i^{\text{sal}} \vee \mathbf{M}^{\text{sim}}, \qquad \mathbf{M}_j = \mathbf{M}_j^{\text{sal}} \vee \mathbf{M}^{\text{sim}}, \quad (4)$$

$$\begin{aligned} \mathbf{W}_{\text{merged}} = \mathbf{M}^{\text{sim}} \odot \frac{|\mathbf{W}_i| + |\mathbf{W}_j|}{2} \\ + (\mathbf{1} - \mathbf{M}^{\text{sim}}) \odot \big(\mathbf{M}_i^{\text{sal}} \odot |\mathbf{W}_i| \\ + \mathbf{M}_j^{\text{sal}} \odot |\mathbf{W}_j|\big). \end{aligned} \quad (5)$$

The merging process is done offline, and $\mathbf{W}_{\text{merged}}$, $\mathbf{M}_i$, $\mathbf{M}_j$, $\mathbf{S}_i$, $\mathbf{S}_j$ are stored. At the inference stage, once an expert is activated, its weights are reconstructed element-wise as

$$\widehat{\mathbf{W}}_i = (-1)^{\mathbf{S}_i} \odot \mathbf{M}_i \odot \mathbf{W}_{\text{merged}}. \qquad (6)$$

**Pair-wise Experts Grouping.** We apply expert merging in a pair-wise manner to balance effectiveness and tractability. Jointly merging $k \geq 3$ experts would introduce a combinatorial masking problem: at each weight index, there are $(2^k - 1)$ possible choices, causing the decision space to grow exponentially with $k$ and making higher-order merging intractable for modern MoE models with many experts. In contrast, pair-wise merging enables closed-form mask construction with linear-time complexity and compact encoding. Beyond tractability, merging more than two experts simultaneously also leads to more aggressive information mixing. Empirically, we observe that higher-order merging results in noticeably larger accuracy degradation under the same overall sparsity. In contrast, pair-wise merging consistently achieves better accuracy-compression trade-offs than higher-order merging strategies, as shown in Appendix A.8.

We consider two strategies for grouping experts: (1) random grouping, (2) search-based grouping. In practice, we find that random grouping performs comparably to search-based alternatives across models and sparsity levels, as shown in Section 5. Therefore, we adopt random grouping by default.

## 3.3. Efficient Inference with Bit-Packing

The sparse expert merging process transforms each pair of experts into a merged expert, 2 sign matrices, and 2 corresponding binary masks, which present a challenge to achieving efficient inference. As shown by Lasby et al. (2025), storing a tensor with 50% unstructured sparsity using Compressed Sparse Row format does not yield memory savings in practice, due to the substantial overhead of index storage. Furthermore, expert computation in MoE LLMs relies on dense matrix multiplication, which is highly optimized on modern GPUs but does not natively support masked operations. Thus, the need to dynamically fetch the appropriate mask during inference introduces nontrivial latency, particularly in the scheduling and memory access of matrix multiplication kernels. To address these limitations, we propose a system-level solution that combines efficient bit-level packing with a high-performance decode-GEMM kernel.

### 3.3.1. OBSERVATION

The BFloat16 data format (1 sign, 8 exponent, and 7 mantissa bits) is standard for LLM inference. Recent analyses (Su et al., 2024; Zhang et al., 2026; Lee et al., 2025a) reveal significant underutilization of the 8-bit exponent range for dense LLMs in BFloat16 format. We found that this

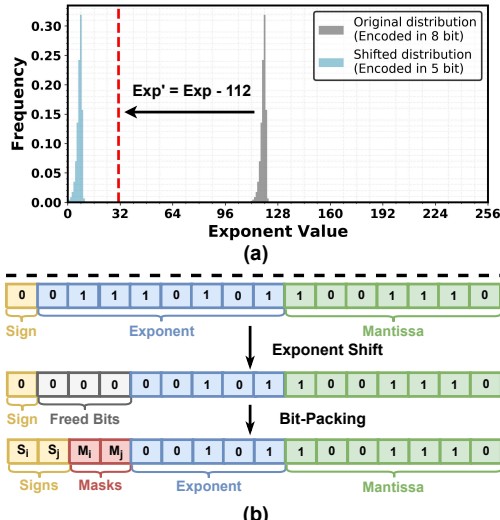

*Figure 4.* Illustration of the bit-packing procedure. (a): the distribution of BFloat16 weight exponents in Mixtral-8x7B. After shifting, the exponents can be encoded in 5 bits. (b): bit-level organization of masks and signs within the packed BFloat16 format.

also applies for MoE LLMs. As shown in Figure 4(a), for Mixtral-8x7B (Jiang et al., 2024) model, the exponent values of expert weights are predominantly concentrated within a narrow range of 112 to 128. We also observe this in other MoE models as shown in Appendix A.3.

### 3.3.2. SYSTEM CO-DESIGN

Leveraging the concentrated exponent distribution, we apply a fixed shift to map all exponents into a 5-bit range, as illustrated in Figure 4. Specifically, any exponent smaller than 112 is rounded up to 112, and then all exponents are shifted down by 112, resulting in values that fall within the range of 0 to 31. The results in Appendix A.4 indicate this transformation incurs no perplexity degradation for existing large MoE models. This is due to the functional equivalence in exponent processing when deriving FP16 weights from a BFloat16 baseline. This shift operation frees 3 bits within the original 8-bit exponent field, which can be used to pack the binary mask bits and a sign bit. Hence, the resultant $\mathbf{W}_{merged}$ is stored in standard BFloat16 format, effectively embedding the masks and signs without additional storage.

We further develop a specialized decode-GEMM kernel in Triton that incorporates on-the-fly decoding to realize the benefit of the above bit-packing idea. The decode operation is shown in Algorithm 1, each weight $W_d[i, j]$ is dynamically decoded from its packed format $W[i, j]$ immediately prior to its use in the multiplication $A[i, j] \times W_d[i, j]$. The efficiency of this approach hinges on its synergistic design. The decoding logic is a computationally trivial, in-place operation that piggybacks on the kernel's existing data-loading path, which is already highly optimized for maximum memory throughput via techniques like warp-level scheduling

---

**Algorithm 1** PuzzleMoE Weight Decoding

---

**Input**: W: Merged weight value, expert_pos: 0 or 1 expert in the merge group
**Output**: $W_d$: Decoded weight value

1: mask_bit ← (W ≫ (13 - expert_pos)) & 1
2: **if** mask_bit==0:
3:     $W_d ← 0$
4: **else**:
5:     sign_bit ← (W ≫ (15 - expert_pos)) & 1
6:     exp ← (W & $0x0F80$) + (112 ≪ 7)
7:     $W_d$ ← (sign_bit ≪ 15) | exp | (W & $0x007F$)
8: $W_d$.view(bfloat16)

---

and coalesced memory access. We eliminate the need for a separate materialization of the decoded matrix in memory, thereby avoiding significant latency and memory access overhead. Moreover, as shown in Figure 5, the proposed decode-GEMM kernel facilitates matrix multiplication for concatenated inputs $A_i, A_j$ by performing a single access to the packed weight tensor to generate the corresponding outputs $O_i, O_j$. While a conventional GEMM kernel necessitates two discrete operations, requiring separate memory transactions for each individual weight tensor. Our design leads to a theoretical $2\times$ latency saving in memory-bound GEMM operation during LLM decoding. We provide a more detailed kernel performance analysis in Appendix A.1.

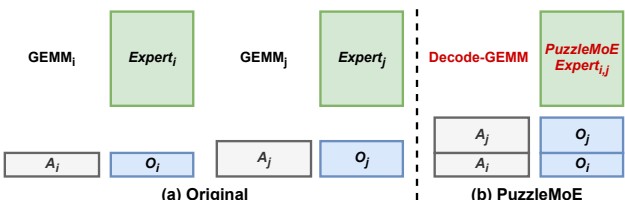

*Figure 5.* Overview of the PuzzleMoE decode-GEMM kernel. (a): the original workflow requires independent access for two experts. (b): PuzzleMoE decode-GEMM workflow only requires accessing the merged expert once to compute the 2 outputs.

# 4. Experiments

## 4.1. Experimental Settings

**Models and Baselines.** We conduct a comprehensive evaluation of PuzzleMoE on four state-of-the-art MoE models: Mixtral-8x7B (Jiang et al., 2024), Deepseek-MoE (Dai et al., 2024), Qwen1.5-MoE-A2.7B (Team, 2024), and Qwen3-MoE-30B-A3B (Yang et al., 2025). All the experiments are conducted on a cluster with 8 RTX5090 GPUs. We follow prior work to evaluate two distinct compression ratios of 25% and 50%, reducing the number of experts to 75% and 50% of the original count, while keeping the other modules unchanged. We compare PuzzleMoE against existing MoE compression methods, including expert dropping methods NAEE (Lu et al., 2024), STUN (Lee et al., 2025b) and

expert merging methods D2 (Gu et al., 2025), HC-SMoE (Chen et al., 2025), Sub-MoE (Li et al., 2025). We also compare with LLM pruning algorithm Wanda (Sun et al., 2024), whose 2:4 semi-structured sparsity is applied exclusively to the experts to ensure a fair comparison [1].

**Benchmarks and Evaluation.** We evaluate the compressed models on language modeling perplexity and zero-shot task performance. Language modeling capabilities are assessed on the WikiText-2 (Merity et al., 2016) with a sequence length of 2048. For downstream tasks, we evaluate zero-shot accuracy across seven common benchmarks: ARC-c (Clark et al., 2018), ARC-e (Clark et al., 2018), HellaSwag (Zellers et al., 2019), PIQA (Bisk et al., 2019), BoolQ (Clark et al., 2019), Winogrande (Sakaguchi et al., 2019), and MMLU (Hendrycks et al., 2021). For all experiments, the calibration dataset has 128 samples, each with a sequence length of 2048 drawn from the C4 dataset (Raffel et al., 2023). We set the similarity threshold $\tau_{\text{sim}} = 0.4$ fixed for all models and tasks, and evaluated 16 different random seeds for expert combination. The results in Table 1, Table 2, and Table 3 for PuzzleMoE are the average result. The detailed results for each seed are shown in Appendix A.9.

## 4.2. Performance on General Tasks

From Table 1, we observe that for Mixtral-8x7B, Puzzle-MoE achieves an average accuracy of 73.2% at 25% sparsity and 72.6% at 50% sparsity, substantially outperforming other baselines. In particular, even under aggressive 50% sparsity, PuzzleMoE still maintains performance very close to the dense model, while existing methods such as HC-SMoE and Sub-MoE suffer significantly larger degradation. Moreover, PuzzleMoE consistently achieves lower perplexity on WikiText compared with other compressed variants, suggesting that the proposed sparse expert merging strategy better preserves the original language modeling capability. For DeepSeek-MoE, PuzzleMoE incurs only minimal accuracy drops of 0.2% and 0.5% at 25% and 50% sparsity, respectively, while maintaining performance consistently superior to prior expert merging approaches. Similarly, for Qwen1.5-MoE, the degradation is limited to only 0.1% and 0.5%, while for Qwen3-MoE, the drops are merely 0.7% and 1.4% under 25% and 50% sparsity. Notably, Puzzle-MoE remains highly competitive on challenging reasoning benchmarks such as MMLU, ARC, and HellaSwag across all model families, demonstrating that the proposed fine-grained sparse expert merging strategy effectively preserves both factual knowledge and reasoning capability after expert compression. Overall, these results highlight the effectiveness and robustness of PuzzleMoE in preserving MoE mod-

---

[1] This is different from the setting in NAEE, where Wanda is applied uniformly across all linear modules in the MoE model, which introduces an unfair comparison. Specifically, the attention module is more sensitive to pruning than the experts module.

*Table 1.* Performance of PuzzleMoE on Mixtral-8x7B, Deepseek-MoE, Qwen1.5-MoE, and Qwen3-MoE on language modeling datasets (perplexity ↓) and 7 common sense reasoning datasets (accuracy ↑).

| Method | Sparsity | Wiki↓ | ARC-c | ARC-e | Hella | Piqa | BoolQ | Wino | MMLU | Avg↑ |
|---|---|---|---|---|---|---|---|---|---|---|
| **Mixtral-8x7B-v0.1** | | | | | | | | | | |
| Vanilla | 0% | 3.84 | 56.7 | 84.1 | 64.9 | 82.4 | 85.4 | 77.2 | 67.9 | 74.1 |
| NAEE | 25% | 5.01 | 52.0 | 82.0 | 61.9 | 80.9 | 84.0 | 75.1 | 58.1 | 70.6 |
| STUN | 25% | - | 52.7 | 81.8 | 60.8 | - | 83.1 | 72.7 | 63.3 | - |
| D2 | 20% | 4.65 | 51.0 | 80.0 | 61.0 | 81.0 | - | 75.0 | - | - |
| HC-SMoE | 25% | 5.31 | 50.3 | 79.3 | 61.3 | 80.7 | 84.9 | 75.4 | 59.4 | 70.2 |
| Sub-MoE | 25% | 5.16 | 49.0 | 80.0 | 62.0 | - | **86.0** | 75.0 | 59.0 | - |
| PuzzleMoE | 25% | **4.10** | **55.3** | **83.2** | **64.2** | **82.1** | 85.4 | **75.5** | **66.8** | **73.2**±0.2 |
| NAEE | 50% | 6.49 | 48.1 | 78.5 | 57.8 | 79.1 | 81.0 | 73.0 | 47.3 | 66.4 |
| D2 | 40% | 5.28 | 47.0 | 78.0 | 57.0 | 78.0 | - | 73.0 | - | - |
| HC-SMoE | 50% | 7.65 | 41.1 | 72.0 | 55.5 | 76.0 | 80.8 | 72.1 | 49.0 | 63.8 |
| Wanda | 2:4 | 5.89 | 48.3 | 78.8 | 58.7 | 79.5 | 79.2 | 74.5 | 62.0 | 68.7 |
| Sub-MoE | 50% | 6.97 | 45.0 | 75.0 | 57.0 | - | 84.0 | 72.0 | 48.0 | - |
| PuzzleMoE | 50% | **4.36** | **53.8** | **82.4** | **63.3** | **81.7** | 85.3 | **75.8** | **65.7** | **72.6**±0.2 |
| **Deepseek-MoE-16b** | | | | | | | | | | |
| Vanilla | 0% | 6.51 | 44.6 | 75.9 | 58.1 | 78.8 | 72.8 | 70.1 | 37.8 | 62.6 |
| D2 | 20% | 6.84 | 41.0 | 74.0 | 55.0 | 76.0 | - | 69.0 | - | - |
| HC-SMoE | 25% | 24.48 | 36.7 | 65.1 | 44.3 | 73.1 | 66.4 | 65.7 | 24.5 | 53.7 |
| Sub-MoE | 25% | 8.48 | 40.0 | 72.0 | 54.0 | - | 73.0 | 70.0 | 27.0 | - |
| PuzzleMoE | 25% | **6.68** | **44.0** | **75.7** | **57.2** | **78.7** | **73.1** | **70.6** | **37.2** | **62.4**±0.3 |
| D2 | 40% | 7.93 | 36.0 | 69.0 | 45.0 | 72.0 | - | 65.0 | - | - |
| Wanda | 2:4 | 8.46 | 37.4 | 71.2 | 51.4 | 76.2 | **75.9** | 69.2 | 31.0 | 58.9 |
| HC-SMoE | 50% | 89.94 | 22.3 | 41.9 | 31.2 | 62.3 | 62.3 | 55.3 | 23.0 | 42.6 |
| Sub-MoE | 50% | 13.71 | 32.0 | 63.0 | 44.0 | - | 68.0 | 65.0 | 22.0 | - |
| PuzzleMoE | 50% | **6.88** | **43.0** | **75.2** | **56.3** | **78.4** | 74.5 | **70.3** | **36.9** | **62.1**±0.4 |
| **Qwen1.5-MoE-A2.7B** | | | | | | | | | | |
| Vanilla | 0% | 7.22 | 41.0 | 73.2 | 58.0 | 80.0 | 79.5 | 68.9 | 61.0 | 65.9 |
| D2 | 20% | 10.91 | 33.6 | 68.3 | 51.0 | 74.6 | 75.1 | 66.8 | 52.4 | 60.3 |
| HC-SMoE | 25% | 11.28 | 34.8 | 67.5 | 50.4 | 74.1 | 74.2 | 66.1 | 51.0 | 59.7 |
| PuzzleMoE | 25% | **7.37** | **40.9** | **73.4** | **57.3** | **79.7** | **79.2** | **69.6** | **60.4** | **65.8**±0.2 |
| D2 | 40% | 27.89 | 27.8 | 47.6 | 42.7 | 68.0 | 66.2 | 60.3 | 37.5 | 50.0 |
| Wanda | 2:4 | 8.81 | 38.7 | 72.2 | 52.6 | 77.5 | 76.2 | 67.8 | 55.8 | 63.0 |
| HC-SMoE | 50% | 78.04 | 23.9 | 41.1 | 31.2 | 60.9 | 56.6 | 56.1 | 23.2 | 41.9 |
| PuzzleMoE | 50% | **7.55** | **40.7** | **73.5** | **56.5** | **79.4** | **78.6** | **69.4** | **60.0** | **65.4**±0.2 |
| **Qwen3-MoE-30B-A3B** | | | | | | | | | | |
| Vanilla | 0% | 8.70 | 52.7 | 79.3 | 59.5 | 79.6 | 88.7 | 70.4 | 77.8 | 72.6 |
| D2 | 20% | 12.78 | 42.1 | 70.6 | 45.1 | 76.6 | 86.2 | 69.1 | 66.2 | 65.1 |
| HC-SMoE | 25% | 14.94 | 40.1 | 70.8 | 48.1 | 71.2 | 82.4 | 63.2 | 60.8 | 62.4 |
| Sub-MoE | 25% | 13.59 | 44.0 | 70.0 | 47.0 | - | 86.0 | 66.0 | 65.0 | - |
| PuzzleMoE | 25% | **9.08** | **51.6** | **78.9** | **58.3** | **79.3** | **88.2** | **70.4** | **76.6** | **71.9**±0.3 |
| D2 | 40% | 31.07 | 35.7 | 65.2 | 43.1 | 70.1 | 83.5 | 60.2 | 47.9 | 58.0 |
| Wanda | 2:4 | 11.77 | 48.2 | 76.1 | 51.1 | 76.3 | 88.0 | **70.5** | 72.1 | 68.9 |
| HC-SMoE | 50% | 42.32 | 34.3 | 60.9 | 44.6 | 69.8 | 80.8 | 54.0 | 39.6 | 54.9 |
| Sub-MoE | 50% | 21.05 | 40.0 | 69.0 | 41.0 | - | 84.0 | 63.0 | 56.0 | - |
| PuzzleMoE | 50% | **9.50** | **51.0** | **78.5** | **57.1** | **78.9** | 88.0 | 70.1 | **75.1** | **71.2**±0.4 |

els' performance after expert merging, particularly under high sparsity regimes where existing methods experience substantial performance degradation.

### 4.3. Performance on Domain-specific Tasks

**Math Reasoning Tasks.** We evaluate PuzzleMoE on domain-specific mathematical reasoning tasks to assess its ability to preserve reasoning performance, as shown in Table 2. PuzzleMoE consistently achieves the best results among all methods, with accuracies of 55.4% and 51.7%

at 25% and 50% sparsity, respectively. Moreover, the effectiveness of NAEE is strongly influenced by the choice of calibration data, performing better with the Math dataset than with C4. In contrast, PuzzleMoE remains robust regardless of calibration datasets as shown in Section 5.

We also evaluate Qwen3-MoE-30B-A3B model (Yang et al., 2025) in reasoning mode on the more challenging reasoning benchmarks (generation-intensive) with results reported in Table 3. Notably, under 25% sparsity, PuzzleMoE effectively preserves the model's reasoning ability after compres-

*Table 2.* 8-shot GSM8K performance on Mixtral-8x7B.

| Sparsity | Method | Calib. data | GSM8K |
|---|---|---|---|
| 0% | - | - | 57.6 |
| 25% | NAEE | C4 | 41.5 |
| | NAEE | Math | 48.7 |
| | PuzzleMoE | C4 | **55.4** |
| 50% | NAEE | C4 | 28.6 |
| | NAEE | Math | 38.8 |
| | Wanda | C4 | 32.2 |
| | PuzzleMoE | C4 | **51.7** |

*Table 3.* Math reasoning benchmarks performance on Qwen3-MoE.

| Method | Math-500 | AIME24 | AIME25 |
|---|---|---|---|
| Baseline (0%) | 97.2 | 83.3 | 72.9 |
| HC-SMoE (25%) | 24.6 | 0.0 | 0.0 |
| PuzzleMoE (25%) | **96.2** | **71.1** | **61.5** |

sion, retaining 99%, 92%, and 84% of accuracy for the three benchmarks, respectively. In sharp contrast, HC-SMoE collapses to 24.6, 0.0, and 0.0 under the same sparsity. This highlights the effectiveness of PuzzleMoE in preserving the reasoning capability of MoE models.

### 4.4. Efficiency Analysis

**Compression Cost.** We measure the time required to compress the MoE models to 50% expert sparsity for PuzzleMoE, D2, HC-SMoE, and NAEE. As shown in Figure 6(a), PuzzleMoE only takes 2 minutes for compressing Mixtral-8x7B, while D2 takes 52 minutes due to the heavy computation cost of SVD operation. For Deepseek-MoE, PuzzleMoE takes 9 minutes for compression, indicating the efficiency of PuzzleMoE on MoE models with a large number of experts. Note that NAEE's reliance on exhaustive search makes the compression time quite expensive. For instance, applying it to DeepSeek-MoE with 64 experts requires $10^{18}$ on the order of forward passes.

**End-to-end Throughput.** While reducing the memory footprint by 50%, PuzzleMoE further yields a substantial increase in end-to-end throughput. We evaluated the throughput of Mixtral-8x7B at a 50% compression ratio, with results illustrated in Figure 6(b). Under a configuration of 16 prefill tokens and 128 decoding tokens across various batch sizes, PuzzleMoE achieves up to a 1.80× speedup compared to the uncompressed baseline. This efficiency gain is primarily driven by our specialized PuzzleMoE decode-GEMM kernel, which delivers a 2× throughput improvement over the standard cuBLAS implementation. For a more comprehensive analysis of kernel benchmarks and end-to-end metrics, please refer to Appendix A.1 and A.2.

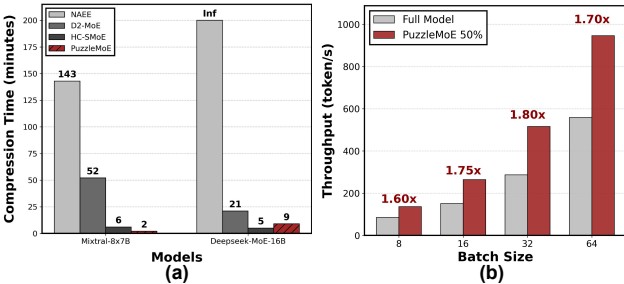

*Figure 6.* Efficiency analysis. (a): compression time comparison across Mixtral and Deepseek-MoE. (b): throughput comparison for Mixtral-8x7B across various batch sizes.

## 5. Ablation Study

**Impact of Calibration Datasets.** We ablate different calibration datasets, and the results are shown in Table 4. It is clear that using C4 or MATH as the calibration dataset doesn't lead to a big variance on performance, emphasizing the robustness of PuzzleMoE.

*Table 4.* Impact of different calibration datasets for PuzzleMoE on Mixtral-8x7B.

| Calib. data | GSM8K | Avg Accuracy |
|---|---|---|
| C4 | 51.7 | 72.6 |
| Math | 51.7 | 72.5 |

**Expert Grouping Strategy.** We compare our random pairwise expert grouping strategy with an evolutionary search-based pairwise grouping strategy. As shown in Table 5, the search-based grouping strategy only leads to a slight performance increase, indicating that PuzzleMoE's effectiveness is hardly influenced by the pairwise grouping strategy.

*Table 5.* Impact of expert grouping strategies at 50% sparsity.

| Model | Method | Wiki | Avg Acc |
|---|---|---|---|
| Mixtral-8x7B | Random | $4.36_{\pm 0.01}$ | $72.6_{\pm 0.2}$ |
| | Searched | 4.35 | 72.9 |
| Deepseek-MoE | Random | $6.88_{\pm 0.01}$ | $62.1_{\pm 0.3}$ |
| | Searched | 6.86 | 62.4 |

**Similarity Threshold $\tau_{\text{sim}}$.** We report perplexity results for PuzzleMoE with different values of $\tau_{\text{sim}}$ as shown in Figure 7(a). It is clear that small values underuse magnitude similarity, while large values merge too aggressively and lead to a substantially higher loss across the two experts. Values of $\tau_{\text{sim}}$ within the range of 0.3 to 0.5 yield the best performance; therefore, we fix $\tau_{\text{sim}} = 0.4$ across models.

**Combining PuzzleMoE with Quantization.** We also investigated the compatibility of PuzzleMoE with quantization. We apply a symmetric group quantization scheme to the merged weights. As shown in Figure 7(b), it achieves a

substantial compression ratio of $4.8\times$, with a minimal accuracy drop of 1.7% for Mixtral-8x7B and 1.0% for Deepseek-MoE compared with full models, demonstrating that our merging strategy can be effectively combined with quantization without significant performance loss. Further details of quantized PuzzleMoE are shown in Appendix A.8.1.

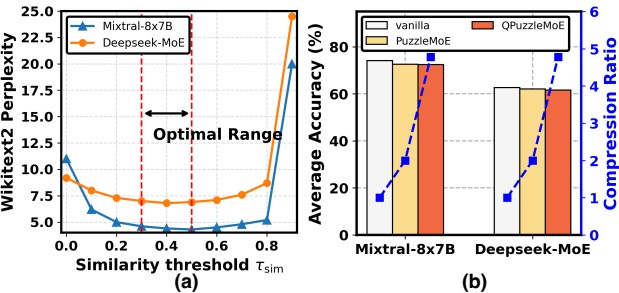

*Figure 7.* (a): Wikitext2 perplexity of Mixtral and Deepseek-MoE under different similarity thresholds. (b): Accuracy performance of combining PuzzleMoE with quantization.

## 6. Conclusion

We introduce PuzzleMoE, the first MoE merging method to enable element-wise merging while achieving both high accuracy and inference speed for large MoE models. Our approach utilizes a fine-grained expert merging strategy guided by dual masks, which enables model compression within minutes while effectively preserving model's capacity. By using bit-packed encoding, PuzzleMoE facilitates efficient decoding on GPUs, offering a practical solution for deploying MoE models in real-world applications.

## Acknowledgments

We sincerely appreciate the anonymous reviewers. Their insightful feedback helps improve the quality of the paper. The work utilized the Delta and DeltaAI system at the National Center for Supercomputing Applications (NCSA) and Jetstream2 at Indiana University through allocation CIS240055 from the Advanced Cyberinfrastructure Coordination Ecosystem: Services & Support (ACCESS) program, which is supported by National Science Foundation grants #2138259, #2138286, #2138307, #2137603, and #2138296. The Delta advanced computing resource is a collaborative effort between the University of Illinois Urbana-Champaign and NCSA, supported by the NSF (award OAC 2005572) and the State of Illinois. UIUC SSAIL Lab is supported by research funding and gift from Google, IBM, Amazon, and AMD, including the Google ML and Systems Junior Faculty Award.

## Impact Statement

This work aims to improve the efficiency and accessibility of Mixture-of-Experts large language models by enabling fine-grained expert merging with minimal performance degradation. By reducing model size, memory footprint, and inference cost, this work lowers the computational barrier to deploying high-capacity language models, potentially broadening access to advanced AI capabilities for researchers and practitioners with limited hardware resources.

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

# A. Appendix

## A.1. Performance of PuzzleMoE Decode-GEMM Kernel

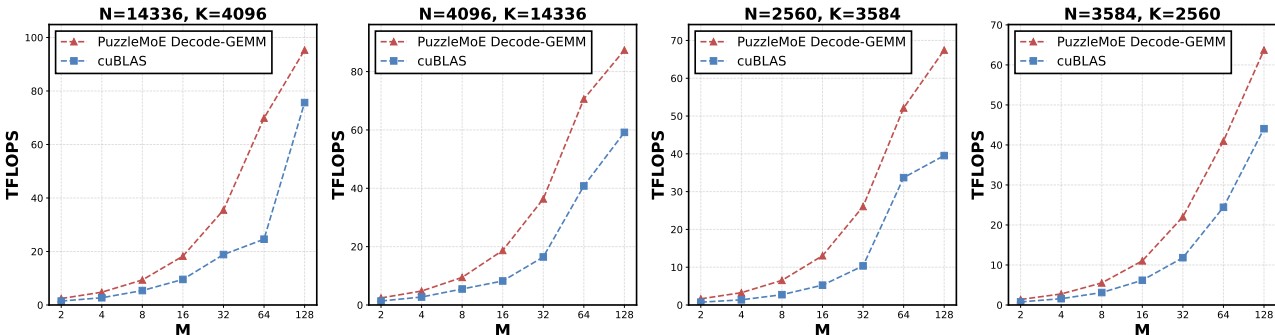

*Figure 8.* Comparison of PuzzleMoE Decode-GEMM and cuBLAS under various matrix shapes on RTX5090 GPU.

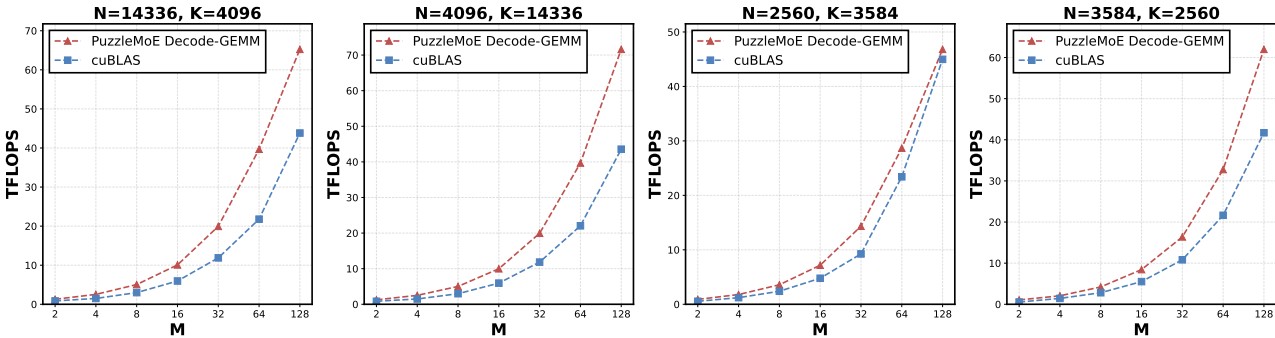

*Figure 9.* Comparison of PuzzleMoE Decode-GEMM and cuBLAS under various matrix shapes on RTX4090 GPU.

We evaluate the performance of our custom kernel on RTX5090 GPU and RTX4090 GPU using Triton Profiler. The benchmarking results, summarized in Figure 8 and Figure 9, span a diverse set of MoE weight matrix dimensions and typical LLM decoding batch sizes. Our PuzzleMoE decode-GEMM kernel consistently achieves approximately a 2× speedup over the optimized cuBLAS baseline. This performance highlights the efficacy of our bit-packing strategy and the optimized memory access patterns of our kernel implementation, which effectively alleviate the memory bottlenecks typically associated with MoE decoding.

## A.2. End-to-end Throughput of PuzzleMoE

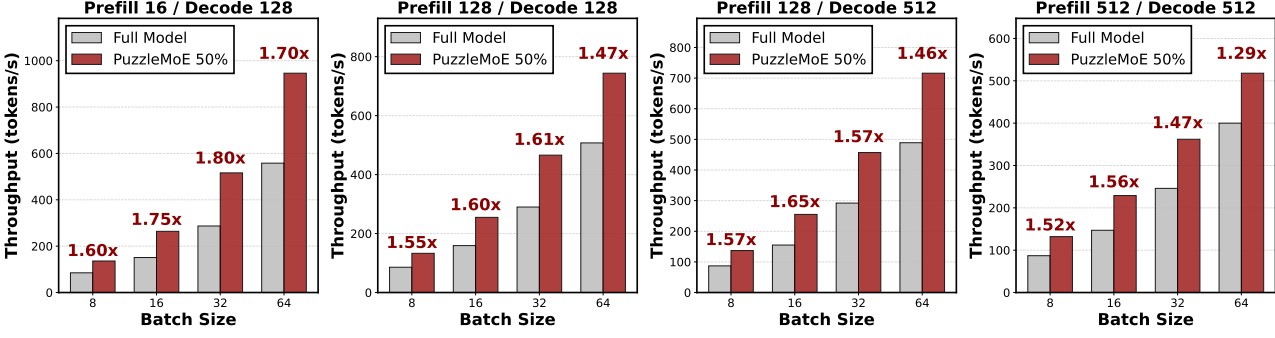

*Figure 10.* End-to-end throughput of PuzzleMoE under various prefill/decode lengths.

We benchmark the end-to-end throughput of PuzzleMoE on a cluster with 8 RTX5090 GPUs at a 50% compression

ratio across various sequence length configurations, specifically (prefill, decode) pairs of $(16, 128)$, $(128, 128)$, $(128, 512)$, and $(512, 512)$. The results, presented in Figure 10, demonstrate that PuzzleMoE achieves up to a $1.8\times$ throughput improvement over the baseline. We observe that the speedup margin diminishes as sequence lengths and batch sizes increase. This trend is attributed to the shifting computational bottleneck: at larger scales, attention operations—which remain uncompressed—constitute a larger fraction of the total execution time, thereby reducing the efficiency gains realized by our optimized MoE kernels.

### A.3. Exponent Distribution Visualization of Different MoE Models

We provide a visualization of the exponent distribution of different MoE models in Figure 11.

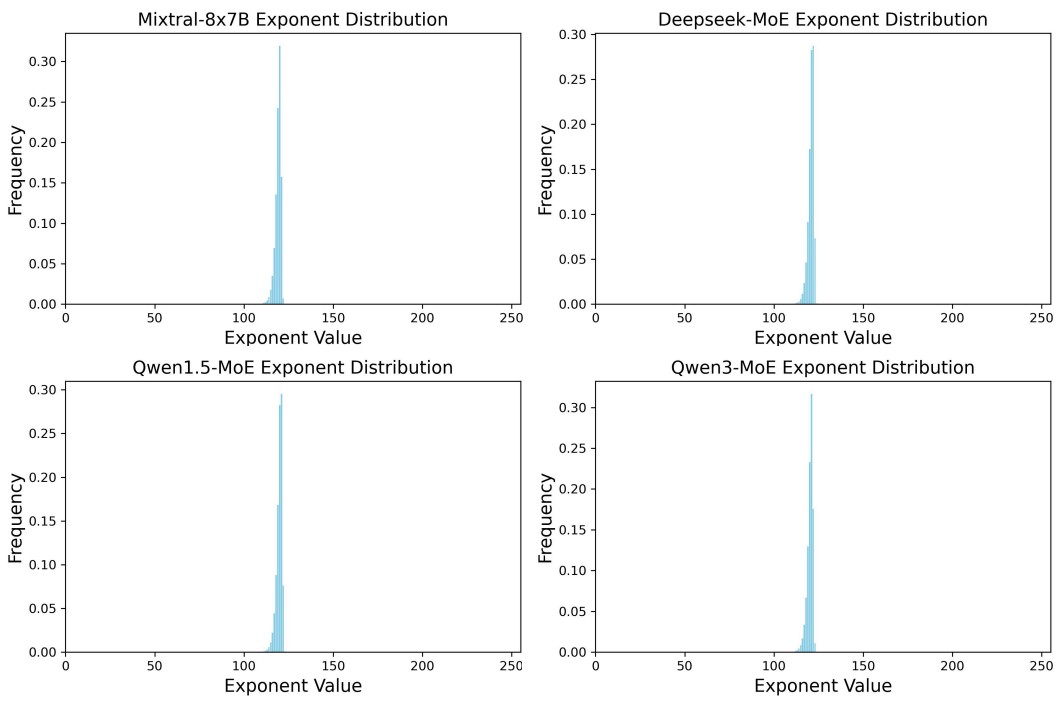

*Figure 11.* Exponent distribution of different MoE models.

### A.4. Impact of Exponent Shifting

Table 6 shows the perplexity results on Wikitext-2 before and after exponent shifting. Exponent shifting incurs no change to model performance.

*Table 6.* Perplexity results on Wikitext-2 before and after exponent shifting.

| Model | Before Shift | After Shift |
|---|---|---|
| Mixtral-8x7B | 4.37 | 4.37 |
| Deepseek-MoE | 6.88 | 6.88 |

### A.5. Visualization of MoE Expert Weight Distribution

We present a visualization of the expert weights (gate_proj parameters) for Mixtral-8x7B, Qwen1.5-MoE, and DeepSeekMoE models in Figure 12, 13, and 14. Our analysis demonstrates that the weight distributions approximately follow a Gaussian distribution. Notably, these distributions are centered at a mean of $\mu = 0$, with a constant standard deviation $\sigma$ across all experts within the same layer of a given model.

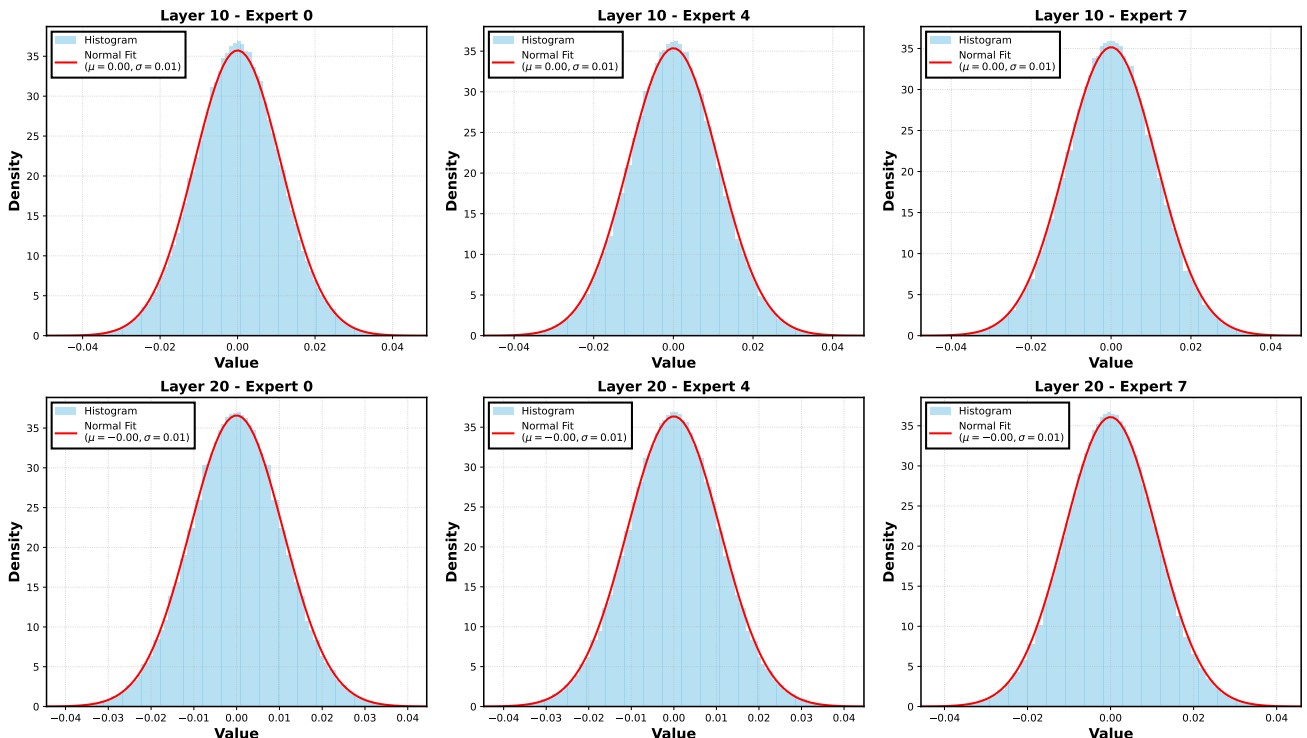

*Figure 12.* Expert weight distribution of Mixtral-8x7B.

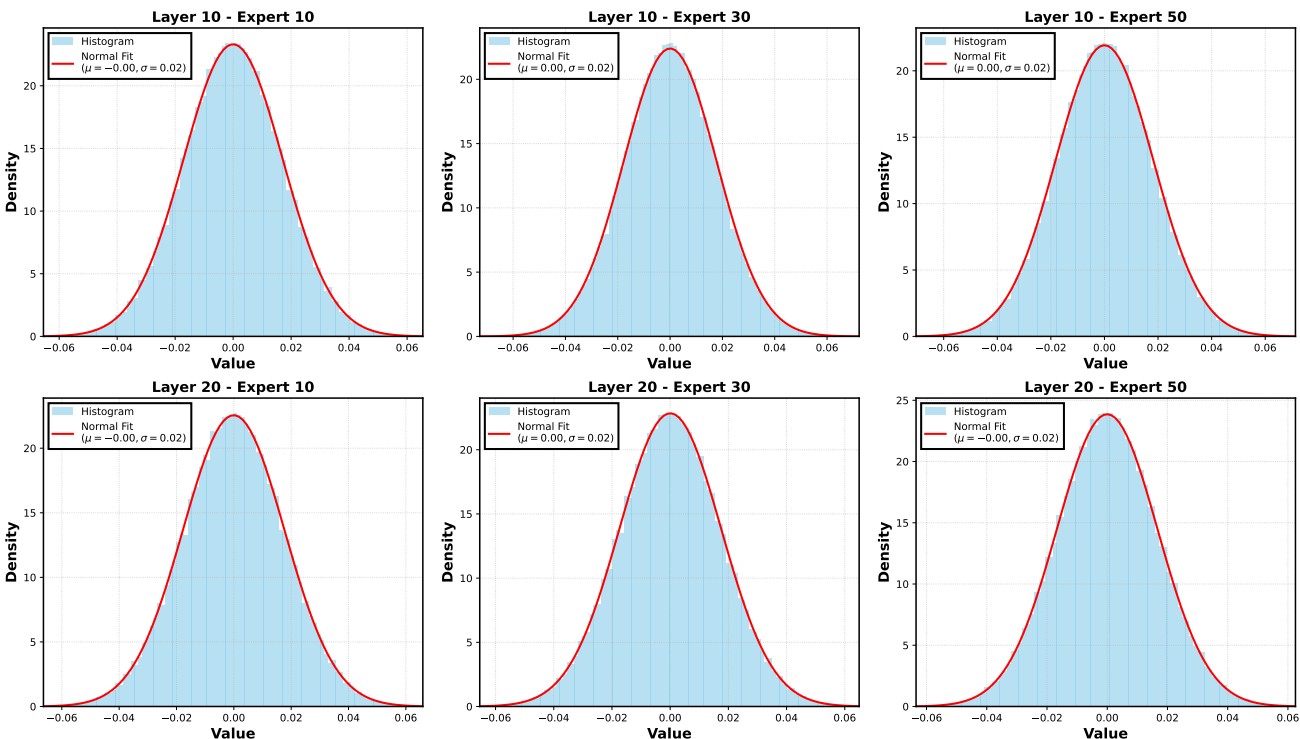

*Figure 13.* Expert weight distribution of Qwen1.5-MoE.

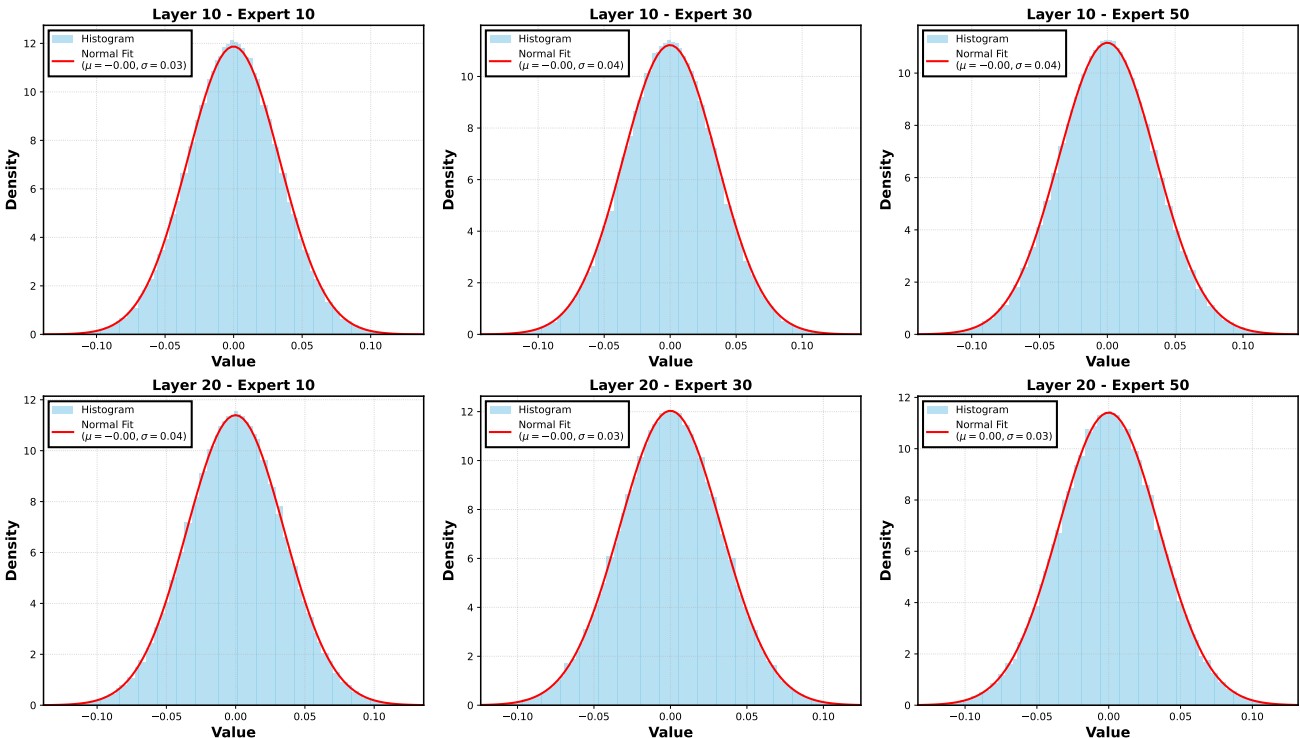

*Figure 14.* Expert weight distribution of Deepseek-MoE.

### A.6. Explanation for Element-wise Similarity in MoE Experts' Weights

We analyze element-level similarity in experts' weights of MoE models. Consistent with previous findings (Kim et al., 2024), the weight distribution of modern LLMs approximates a normal distribution with a zero mean. To quantify inter-expert similarity, we computed the Pearson correlation coefficient between tensors in different pairs of experts within each layer, then we averaged the result for the whole model. As shown in Table 7, for Qwen1.5-MoE and Deepseek-MoE, the correlation is negligible. Mixtral-8x7B exhibits a higher correlation, indicating stronger dependencies between its expert weights. Consequently, for Qwen1.5-MoE and Deepseek-MoE, the values within their respective tensors can be treated as independent variables.

*Table 7.* Average Pearson correlation of expert weights.

| Model | Average correlation |
|---|---|
| Mixtral-8x7B | 0.2612 |
| Deepseek-MoE | 0.0006 |
| Qwen1.5-MoE | 0.0000 |

Based on this setting, we can view the process of selecting 2 values $w_i, w_j$ of the corresponding positions of the 2 weights $\mathbf{W}_i, \mathbf{W}_j$ as a sampling process. We define and solve the problem of calculating the proportion of similar values as follows:

We consider independent Gaussian random variables

$$w_i \sim N(0, \sigma_j^2), \quad w_j \sim N(0, \sigma_j^2),$$

We aim to compute

$$P\left( \frac{||w_i| - |w_j||}{|w_i| + |w_j|} < \tau_{\text{sim}} \right), \qquad 0 < \tau_{\text{sim}} < 1.$$

Since $|w_i|, |w_j| \geq 0$, we have

$$\frac{||w_i| - |w_j||}{|w_i| + |w_j|} < \tau_{\text{sim}} \iff ||w_i| - |w_j|| < \tau_{\text{sim}}(|w_i| + |w_j|) \iff \frac{1 - \tau_{\text{sim}}}{1 + \tau_{\text{sim}}} < \frac{|w_i|}{|w_j|} < \frac{1 + \tau_{\text{sim}}}{1 - \tau_{\text{sim}}}.$$

Define the ratio

$$R = \frac{|w_i|}{|w_j|}.$$

The densities of $|w_i|$ and $|w_j|$ (half-normal distributions) are given by

$$f_1(|w_i|) = \frac{\sqrt{2}}{\sigma_i \sqrt{\pi}} e^{-|w_i|^2/(2\sigma_i^2)},$$

$$f_2(|w_j|) = \frac{\sqrt{2}}{\sigma_j \sqrt{\pi}} e^{-|w_j|^2/(2\sigma_j^2)}.$$

For $R = |w_i|/|w_j|$, its pdf is

$$f_R(r) = \int_0^\infty |w_j| \, f_1(r|w_j|) \, f_2(|w_j|) \, d(|w_j|) = \frac{2\sigma_i \sigma_j}{\pi(\sigma_i^2 + \sigma_j^2 r^2)}, \quad r \geq 0.$$

Let

$$a = \frac{1 - \tau_{\text{sim}}}{1 + \tau_{\text{sim}}}, \qquad b = \frac{1 + \tau_{\text{sim}}}{1 - \tau_{\text{sim}}}.$$

Then

$$P = \int_a^b f_R(r) \, dr = \frac{2\sigma_i \sigma_j}{\pi} \int_a^b \frac{dr}{\sigma_i^2 + \sigma_j^2 r^2} = \frac{2}{\pi} \left[ \arctan\left(\frac{\sigma_j r}{\sigma_i}\right) \right]_a^b.$$

Hence, the final result is

$$P\left( \frac{||w_i| - |w_j||}{|w_i| + |w_j|} < \tau_{\text{sim}} \right) = \frac{2}{\pi} \left[ \arctan\left(\frac{\sigma_j}{\sigma_i} \frac{1 + \tau_{\text{sim}}}{1 - \tau_{\text{sim}}}\right) - \arctan\left(\frac{\sigma_j}{\sigma_i} \frac{1 - \tau_{\text{sim}}}{1 + \tau_{\text{sim}}}\right) \right].$$

In the case $\sigma_i \approx \sigma_j$, which typically holds for most expert weights in MoE models as shown in A.5, we obtain

$$P\left( \frac{||w_i| - |w_j||}{|w_i| + |w_j|} < \tau_{\text{sim}} \right) = \frac{2}{\pi} \left[ \arctan\left(\frac{1 + \tau_{\text{sim}}}{1 - \tau_{\text{sim}}}\right) - \arctan\left(\frac{1 - \tau_{\text{sim}}}{1 + \tau_{\text{sim}}}\right) \right].$$

The theoretical curve and empirical data from the three MoE models are plotted in Figure 15. The observed results for Qwen1.5-MoE and Deepseek-MoE align closely with the theoretical predictions. The data for Mixtral-8x7B shows a slight deviation, which is attributable to its violation of the independent distribution assumption. This analysis provides a clear explanation for the existence of fine-grained, element-wise similarity in expert weights and reinforces the generalizability and explainability of our method.

### A.7. Performance on Coding Tasks

We further investigate the knowledge preservation ability of PuzzleMoE on coding tasks. Specifically, we compare different expert merging algorithms on Qwen1.5-MoE-A2.7B and evaluate them on HumanEval (Chen et al., 2021). As shown in Table 8, PuzzleMoE preserves code-generation performance much better than HC-SMoE under the same sparsity levels: at 25% sparsity, PuzzleMoE attains a Pass@1 of 45.1 with only a small drop from the 49.4 dense baseline, while HC-SMoE collapses to 11.0; at 50% sparsity, PuzzleMoE still achieves 39.6, whereas HC-SMoE completely fails with 0.0 Pass@1. These results demonstrate that PuzzleMoE maintains strong coding ability even at high sparsity, while prior MoE merging methods severely degrade model reliability.

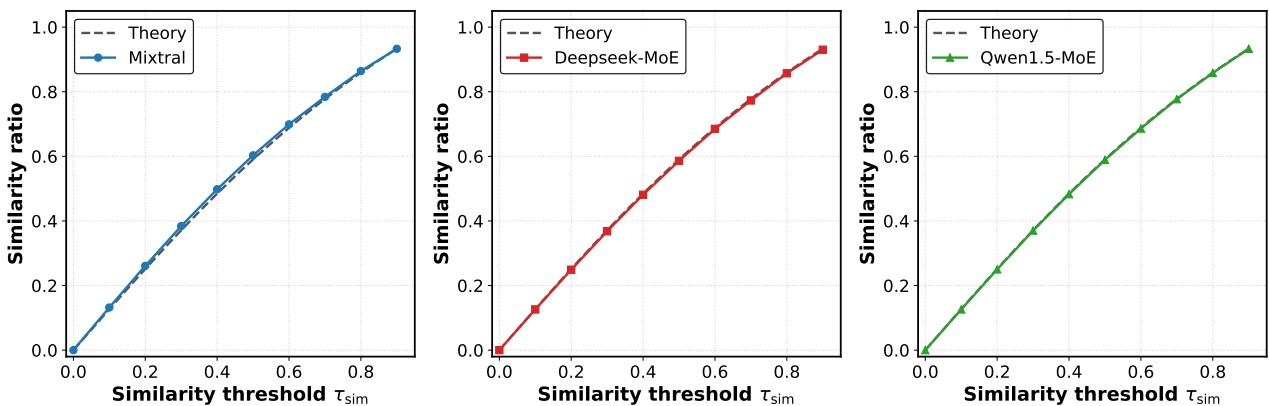

*Figure 15.* Relation between similarity ratio and similarity threshold $\tau_{\mathrm{sim}}$.

*Table 8.* HumanEval performance on Qwen1.5-MoE.

| Model | HumanEval Pass@1 |
|---|---|
| Baseline (0%) | 49.4 |
| HC-SMoE (25%) | 11.0 |
| PuzzleMoE (25%) | **45.1** |
| HC-SMoE (50%) | 0.0 |
| PuzzleMoE (50%) | **39.6** |

## A.8. More Ablation Study

### A.8.1. COMBINING PUZZLEMOE WITH QUANTIZATION

We provide a detailed description of combining PuzzleMoE with quantization, as depicted in Figure 16. After the expert merging stage, the resultant floating-point weights are subjected to uniform quantization with a group size of 128. We employ a symmetric group quantization scheme, which obviates the need for a zero point. The final quantized values are stored alongside their corresponding sign and mask bits. The average bit width is subsequently determined by the following equation:

$$\text{Avg bitwidth} = (\underbrace{2}_{\text{Sign}} + \underbrace{1.58}_{\text{Compressed mask}} + \underbrace{3}_{\text{Quantize bitwidth}} + \underbrace{0.125}_{\text{Per-group scale}})/2 = 3.35$$

Each compressed mask belongs to the set $\{01,10,11\}$, as for each position, the merged weight either belongs to $W_i$, $W_j$, or both. Consequently, it requires only $\log_2 3 \approx 1.58$ bits for representation.

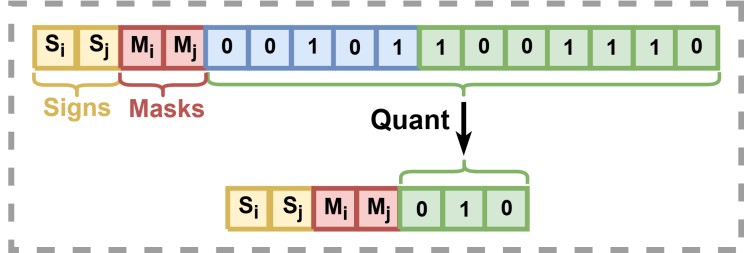

*Figure 16.* Combining PuzzleMoE with quantization.

We target an 80% compression ratio, as this represents a high level of compression without a substantial loss in model performance. Accordingly, we apply 3-bit quantization to the merged weights within the PuzzleMoE framework at 50% sparsity. To evaluate our approach, we compare the post-quantization performance of the expert modules using 3-bit AWQ (Lin et al., 2024) and NAEE/HC-SMoE with 25% sparsity + 4-bit AWQ. The results are shown in Table 9.

On Mixtral-8x7B, AWQ attains the highest average accuracy of 73.0 at 3.25 bits, while PuzzleMoE achieves a comparable 72.4 accuracy with a similar bitwidth of 3.35. Compared to NAEE+AWQ and HC-SMoE+AWQ, which both reach only 70.1 average accuracy, PuzzleMoE provides a clear gain of more than 2 points, indicating that expert merging via PuzzleMoE preserves quantization performance better than prior MoE compression schemes. On Deepseek-MoE, PuzzleMoE further improves both Wiki and average accuracy at a bitwidth of 3.35, whereas HC-SMoE+AWQ suffers a substantial drop to 53.3 average accuracy. Overall, these results suggest that PuzzleMoE can be seamlessly combined with low-bit quantization, delivering robust accuracy improvements over NAEE and HC-SMoE at only a negligible increase in effective bitwidth.

*Table 9.* Performance of combining PuzzleMoE with quantization.

| Model | Method | Avg bitwidth | Wiki | Avg Accuracy |
|---|---|---|---|---|
| Mixtral-8x7B | NAEE+AWQ | 3.19 | 5.10 | 70.1 |
| | AWQ | 3.25 | 4.41 | 73.0 |
| | HC-SMoE+AWQ | 3.30 | 5.39 | 70.1 |
| | PuzzleMoE | 3.35 | 4.50 | 72.4 |
| Deepseek-MoE | AWQ | 3.25 | 6.87 | 61.2 |
| | HC-SMoE+AWQ | 3.30 | 26.7 | 53.3 |
| | PuzzleMoE | 3.35 | 7.05 | 61.6 |

### A.8.2. IMPACT OF NUMBER OF EXPERTS TO MERGE

We ablate the number of experts to merge in PuzzleMoE. As shown in Table 10, at a 50% sparsity level, increasing the merge group size from 2 to 3 experts leads to degraded performance, as reflected by higher perplexity.

*Table 10.* Impact of different numbers of experts to merge with 50% sparsity.

| Model | Merge Number | Wiki |
|---|---|---|
| Mixtral-8x7B | 2 | 4.36 |
| | 3 | 5.22 |
| Deepseek-MoE | 2 | 6.88 |
| | 3 | 7.75 |

### A.8.3. IMPACT OF SALIENCY METRICS

To further investigate the effectiveness of the Wanda saliency metric, we evaluate several MoE-aware variants that incorporate router information to different extents: W ($|W|$), WG ($|W| \cdot \|gate\|_2$), WA ($|W| \cdot \|X\|_2$, identical to the original Wanda metric), and WAG ($|W| \cdot \|X \cdot gate\|_2$), where $W$ denotes the weight matrix, $X$ the activation, and $gate$ the router output.

As shown in Table 11, the activation-based variants (WA and WAG) achieve very similar overall performance and consistently match or slightly outperform the weight-only variants (W and WG). In contrast, adding gating scores on top of weight-only saliency (WG vs. W) brings almost no benefit. This indicates that the main performance gains stem from incorporating activation information into the saliency metric, which better preserves expert specialization, while additional router-gating signals provide no significant improvements.

*Table 11.* Performance of PuzzleMoE with different MoE-aware saliency metrics at 50% sparsity.

| Model | Metric | Wiki | Avg |
|---|---|---|---|
| Mixtral-8x7B | $|W|$ | 4.38 | 72.6 |
| | $|W| \cdot \|gate\|_2$ | 4.38 | 72.6 |
| | $|W| \cdot \|X\|_2$ | **4.36** | **72.6** |
| | $|W| \cdot \|X \cdot gate\|_2$ | 4.36 | 72.6 |
| DeepSeek-MoE | $|W|$ | 6.98 | 61.8 |
| | $|W| \cdot \|gate\|_2$ | 6.96 | 61.7 |
| | $|W| \cdot \|X\|_2$ | **6.88** | **62.1** |
| | $|W| \cdot \|X \cdot gate\|_2$ | 6.89 | 62.1 |

*Table 12.* Performance of PuzzleMoE and HC-SMoE on Deepseek-MoE and Qwen3-MoE across 7 reasoning benchmarks (accuracy ↑).

| Method | Sparsity | ARC-c | ARC-e | Hella | Piqa | BoolQ | Wino | MMLU | Avg↑ |
|--------|----------|-------|-------|-------|------|-------|------|------|------|
| | | | | **Deepseek-MoE-16b** | | | | | |
| HC-SMoE | 25% | 36.7 | 65.1 | 44.3 | 73.1 | 66.4 | 65.7 | 24.5 | 53.7 |
| PuzzleMoE | 25% | **44.0** | **75.7** | **57.2** | **78.7** | **73.1** | **70.6** | **37.2** | **62.4** |
| HC-SMoE | 37.5% | 31.2 | 58.9 | 41.3 | 72.0 | 65.1 | 63.4 | 24.7 | 50.9 |
| PuzzleMoE | 37.5% | **44.2** | **75.3** | **56.2** | **78.7** | **73.3** | **69.9** | **36.9** | **62.1** |
| HC-SMoE | 50% | 22.3 | 41.9 | 31.2 | 62.3 | 62.3 | 55.3 | 23.0 | 42.6 |
| PuzzleMoE | 50% | **43.0** | **75.2** | **56.3** | **78.4** | **74.5** | **70.3** | **36.9** | **62.1** |
| HC-SMoE | 66.7% | 20.5 | 38.6 | 26.7 | 60.5 | 62.0 | 54.3 | 24.5 | 41.0 |
| PuzzleMoE | 66.7% | **36.9** | **70.1** | **49.8** | **75.6** | **73.1** | **70.1** | **29.6** | **57.9** |
| HC-SMoE | 75% | 19.9 | 27.1 | 26.3 | 54.3 | 59.8 | 50.8 | 23.6 | 35.7 |
| PuzzleMoE | 75% | **33.4** | **65.3** | **54.1** | **72.2** | **62.2** | **66.4** | **24.9** | **54.1** |
| | | | | **Qwen3-MoE-30B-A3B** | | | | | |
| HC-SMoE | 25% | 40.1 | 70.8 | 48.1 | 71.2 | 82.4 | 63.2 | 60.8 | 62.4 |
| PuzzleMoE | 25% | **51.6** | **78.9** | **58.3** | **79.3** | **88.2** | **70.4** | **76.6** | **71.9** |
| HC-SMoE | 37.5% | 37.6 | 67.8 | 46.2 | 71.0 | 81.2 | 60.0 | 53.7 | 59.6 |
| PuzzleMoE | 37.5% | **51.4** | **78.5** | **57.4** | **78.9** | **87.2** | **70.5** | **75.8** | **71.4** |
| HC-SMoE | 50% | 34.3 | 60.9 | 44.6 | 69.8 | 80.8 | 54.0 | 39.6 | 54.9 |
| PuzzleMoE | 50% | **51.0** | **78.5** | **57.1** | **78.9** | **88.0** | **70.1** | **75.1** | **71.2** |
| HC-SMoE | 66.7% | 28.9 | 40.2 | 35.7 | 60.1 | 78.6 | 53.0 | 33.4 | 47.1 |
| PuzzleMoE | 66.7% | **43.7** | **74.4** | **53.9** | **76.2** | **86.9** | **69.9** | **63.8** | **67.0** |
| HC-SMoE | 75% | 25.6 | 31.2 | 28.8 | 54.3 | 76.4 | 52.5 | 27.2 | 42.3 |
| PuzzleMoE | 75% | **33.7** | **63.2** | **46.8** | **72.5** | **80.1** | **69.1** | **48.1** | **59.1** |

### A.8.4. EXTENDING PUZZLEMOE TO HIGHER SPARSITY LEVELS

The current bit-packing design is indeed tailored to 2-to-1 merging. However, this does not constrain the dual-mask merging algorithm itself. If we store masks and signs explicitly without packing them into BFloat16, the same dual-mask merging algorithm can, in principle, support arbitrary compression ratios (e.g., 3-to-1, 4-to-1). In other words, the bit-packing scheme is introduced solely to reduce the memory footprint and make high-sparsity operation practical; it is not a limitation of the merging algorithm or of the achievable sparsity.

We further evaluate higher sparsity levels of 66.7% (merging three experts into one) and 75% (merging four experts into one) for both Deepseek-MoE-16b and Qwen3-MoE-30B-A3B. The evaluation follows the same benchmarks used in the submission (ARC-C/E, HellaSwag, PIQA, BoolQ, WinoGrande, and MMLU). The following tables report average results across these benchmarks. Under highly sparse settings, PuzzleMoE consistently outperforms the strong HC-SMoE baseline. For example, for Deepseek-MoE-16b at 75% sparsity, PuzzleMoE improves the average score from 35.7 to 54.1, demonstrating its strong robustness even under aggressive compression compared to the existing work.

### A.9. Detailed Results of PuzzleMoE with Different Seeds

The detailed accuracy results of PuzzleMoE with different random seeds on Mixtral-8x7B, Deepseek-MoE, Qwen1.5-MoE, and Qwen3-MoE are shown in Table 13, 14, 15, 16. Different seeds don't lead to a significant difference in accuracy performance for both 25% sparsity and 50% sparsity, indicating the robustness of PuzzleMoE to different expert grouping choices.

*Table 13.* Zero-shot results of PuzzleMoE on Mixtral-8x7B under 25% and 50% sparsity with different seeds.

| Sparsity | Seed | Wiki | ARC-c | ARC-e | Hella | Piqa | BoolQ | Wino | MMLU | Avg |
|---|---|---|---|---|---|---|---|---|---|---|
| 0% | - | 3.84 | 56.7 | 84.1 | 64.9 | 82.4 | 85.4 | 77.2 | 67.9 | 74.1 |
| 25% | 1 | 4.09 | 55.4 | 83.4 | 64.1 | 82.2 | 85.5 | 75.3 | 66.6 | 73.2 |
| | 2 | 4.11 | 56.2 | 83.1 | 65.2 | 81.8 | 84.4 | 74.9 | 66.6 | 73.2 |
| | 3 | 4.11 | 55.1 | 82.8 | 64.0 | 82.2 | 85.6 | 75.4 | 67.1 | 73.2 |
| | 4 | 4.10 | 55.0 | 82.7 | 63.9 | 81.7 | 85.9 | 75.0 | 67.1 | 73.0 |
| | 5 | 4.10 | 54.8 | 83.1 | 64.0 | 81.7 | 85.4 | 75.4 | 66.6 | 73.0 |
| | 6 | 4.10 | 53.7 | 82.5 | 63.9 | 81.9 | 85.9 | 75.4 | 67.0 | 72.9 |
| | 7 | 4.09 | 56.3 | 83.4 | 64.1 | 82.5 | 85.4 | 74.8 | 66.7 | 73.3 |
| | 8 | 4.09 | 56.6 | 83.5 | 64.4 | 81.9 | 85.3 | 76.6 | 66.7 | 73.6 |
| | 9 | 4.09 | 55.2 | 83.5 | 64.2 | 82.2 | 85.3 | 75.9 | 66.6 | 73.3 |
| | 10 | 4.11 | 55.6 | 83.1 | 64.2 | 81.7 | 85.8 | 76.1 | 67.0 | 73.4 |
| | 11 | 4.10 | 55.0 | 82.8 | 64.2 | 82.3 | 85.3 | 75.0 | 66.4 | 73.0 |
| | 12 | 4.13 | 55.5 | 83.2 | 64.5 | 82.4 | 85.5 | 75.4 | 67.3 | 73.4 |
| | 13 | 4.10 | 54.6 | 83.3 | 64.1 | 82.1 | 85.8 | 75.5 | 66.5 | 73.1 |
| | 14 | 4.10 | 54.7 | 83.4 | 64.2 | 82.5 | 85.6 | 76.0 | 67.2 | 73.4 |
| | 15 | 4.09 | 56.5 | 83.6 | 64.3 | 81.8 | 84.1 | 76.5 | 66.9 | 73.4 |
| | 16 | 4.09 | 54.8 | 83.1 | 64.2 | 82.3 | 85.7 | 74.6 | 67.1 | 73.1 |
| | avg | 4.10 | 55.3 | 83.2 | 64.2 | 82.1 | 85.4 | 75.5 | 66.8 | 73.2 |
| | std | 0.01 | 0.8 | 0.3 | 0.3 | 0.3 | 0.5 | 0.6 | 0.3 | 0.2 |
| 50% | 1 | 4.36 | 53.1 | 82.5 | 63.3 | 82.1 | 84.6 | 76.4 | 65.8 | 72.5 |
| | 2 | 4.35 | 53.5 | 82.3 | 63.4 | 81.5 | 84.2 | 74.9 | 65.6 | 72.2 |
| | 3 | 4.35 | 53.3 | 82.2 | 63.4 | 81.6 | 86.5 | 75.9 | 66.0 | 72.7 |
| | 4 | 4.36 | 53.6 | 82.2 | 63.2 | 81.6 | 86.5 | 75.9 | 65.7 | 72.7 |
| | 5 | 4.36 | 54.3 | 82.3 | 63.2 | 81.5 | 85.1 | 75.5 | 65.5 | 72.5 |
| | 6 | 4.37 | 53.8 | 82.2 | 63.2 | 81.6 | 85.6 | 77.0 | 65.8 | 72.7 |
| | 7 | 4.37 | 54.0 | 82.7 | 63.4 | 82.1 | 85.8 | 75.9 | 65.8 | 72.8 |
| | 8 | 4.35 | 53.6 | 82.5 | 63.1 | 81.0 | 85.5 | 76.7 | 65.6 | 72.6 |
| | 9 | 4.35 | 54.4 | 82.5 | 63.2 | 81.8 | 85.2 | 75.8 | 65.4 | 72.6 |
| | 10 | 4.35 | 53.8 | 82.0 | 63.3 | 81.6 | 85.1 | 75.3 | 65.4 | 72.4 |
| | 11 | 4.35 | 53.6 | 82.3 | 63.4 | 81.6 | 83.6 | 75.5 | 65.9 | 72.3 |
| | 12 | 4.37 | 54.0 | 82.3 | 63.5 | 82.2 | 85.5 | 75.1 | 66.0 | 72.7 |
| | 13 | 4.36 | 53.9 | 82.5 | 63.3 | 81.8 | 85.7 | 76.2 | 65.1 | 72.6 |
| | 14 | 4.35 | 53.4 | 82.9 | 63.4 | 81.8 | 86.6 | 74.9 | 65.8 | 72.9 |
| | 15 | 4.37 | 53.7 | 82.8 | 63.3 | 81.1 | 84.2 | 75.9 | 66.2 | 72.5 |
| | 16 | 4.35 | 54.0 | 82.4 | 63.2 | 82.0 | 84.3 | 75.1 | 65.9 | 72.4 |
| | avg | 4.36 | 53.8 | 82.4 | 63.3 | 81.7 | 85.3 | 75.8 | 65.8 | 72.6 |
| | std | 0.01 | 0.3 | 0.2 | 0.1 | 0.3 | 0.9 | 0.6 | 0.3 | 0.2 |

*Table 14.* Zero-shot results of PuzzleMoE on Deepseek-MoE under 25% and 50% sparsity with different seeds.

| Sparsity | Seed | Wiki | ARC-c | ARC-e | Hella | Piqa | BoolQ | Wino | MMLU | Avg |
|---|---|---|---|---|---|---|---|---|---|---|
| 0% | - | 6.51 | 44.6 | 75.9 | 58.1 | 78.8 | 72.8 | 70.1 | 37.8 | 62.6 |
| 25% | 1 | 6.69 | 44.8 | 75.8 | 57.2 | 78.4 | 72.4 | 70.6 | 37.1 | 62.3 |
| | 2 | 6.67 | 43.9 | 75.9 | 57.3 | 78.2 | 70.7 | 70.5 | 35.4 | 61.7 |
| | 3 | 6.67 | 43.3 | 75.1 | 57.3 | 78.8 | 71.8 | 70.2 | 37.3 | 62.0 |
| | 4 | 6.68 | 44.4 | 75.7 | 57.5 | 78.9 | 71.0 | 70.6 | 37.2 | 62.2 |
| | 5 | 6.67 | 43.3 | 75.7 | 57.3 | 78.3 | 74.0 | 70.5 | 37.9 | 62.4 |
| | 6 | 6.67 | 43.8 | 75.6 | 57.1 | 78.7 | 72.9 | 71.0 | 38.0 | 62.4 |
| | 7 | 6.68 | 43.9 | 75.4 | 56.9 | 79.1 | 73.6 | 70.6 | 36.9 | 62.3 |
| | 8 | 6.70 | 44.1 | 75.6 | 56.8 | 78.8 | 75.1 | 70.7 | 36.2 | 62.5 |
| | 9 | 6.68 | 45.1 | 76.1 | 57.0 | 78.8 | 73.5 | 69.7 | 38.2 | 62.6 |
| | 10 | 6.68 | 44.2 | 75.4 | 57.3 | 78.4 | 73.7 | 69.9 | 37.0 | 62.3 |
| | 11 | 6.67 | 43.9 | 75.5 | 57.4 | 78.6 | 72.1 | 70.5 | 37.8 | 62.3 |
| | 12 | 6.66 | 44.0 | 76.2 | 57.2 | 78.3 | 72.5 | 70.9 | 36.2 | 62.2 |
| | 13 | 6.70 | 43.9 | 75.7 | 57.2 | 78.3 | 74.8 | 70.5 | 37.8 | 62.6 |
| | 14 | 6.70 | 43.4 | 76.4 | 57.4 | 79.2 | 73.8 | 70.6 | 36.2 | 62.4 |
| | 15 | 6.67 | 44.4 | 76.3 | 57.4 | 78.9 | 74.6 | 71.1 | 37.5 | 62.9 |
| | 16 | 6.67 | 43.9 | 75.4 | 57.4 | 78.9 | 73.3 | 70.9 | 39.2 | 62.7 |
| | avg | 6.68 | 44.0 | 75.7 | 57.2 | 78.7 | 73.1 | 70.6 | 37.2 | 62.4 |
| | std | 0.01 | 0.5 | 0.4 | 0.2 | 0.3 | 1.3 | 0.4 | 0.9 | 0.3 |
| 50% | 1 | 6.90 | 43.3 | 75.6 | 56.3 | 77.8 | 74.9 | 69.3 | 36.1 | 61.9 |
| | 2 | 6.89 | 42.8 | 75.9 | 56.3 | 78.3 | 73.4 | 70.5 | 35.5 | 61.8 |
| | 3 | 6.88 | 42.2 | 74.5 | 56.5 | 78.3 | 74.6 | 70.0 | 36.1 | 61.7 |
| | 4 | 6.87 | 43.1 | 75.8 | 56.7 | 78.5 | 75.0 | 69.5 | 37.2 | 62.3 |
| | 5 | 6.87 | 42.5 | 75.2 | 55.8 | 78.5 | 76.0 | 70.0 | 38.1 | 62.3 |
| | 6 | 6.88 | 41.9 | 74.6 | 56.6 | 78.4 | 74.7 | 71.2 | 37.2 | 62.1 |
| | 7 | 6.88 | 42.8 | 74.6 | 56.5 | 78.6 | 75.1 | 70.9 | 37.9 | 62.3 |
| | 8 | 6.89 | 43.5 | 75.1 | 56.5 | 78.2 | 73.7 | 69.9 | 36.7 | 61.9 |
| | 9 | 6.88 | 43.9 | 75.8 | 56.5 | 78.9 | 74.7 | 70.9 | 37.9 | 62.7 |
| | 10 | 6.89 | 43.3 | 74.8 | 56.5 | 78.7 | 74.9 | 71.0 | 36.9 | 62.3 |
| | 11 | 6.87 | 44.1 | 75.5 | 55.6 | 78.6 | 73.2 | 69.7 | 37.6 | 62.0 |
| | 12 | 6.85 | 43.1 | 74.8 | 56.4 | 77.8 | 74.3 | 69.6 | 36.2 | 61.7 |
| | 13 | 6.89 | 43.1 | 74.8 | 56.3 | 78.2 | 76.3 | 71.0 | 37.2 | 62.4 |
| | 14 | 6.89 | 41.5 | 75.9 | 55.7 | 78.2 | 72.4 | 70.6 | 34.3 | 61.2 |
| | 15 | 6.88 | 43.5 | 75.0 | 56.3 | 78.7 | 74.1 | 70.6 | 36.8 | 62.1 |
| | 16 | 6.87 | 42.7 | 74.7 | 56.5 | 78.8 | 75.2 | 69.9 | 38.6 | 62.3 |
| | avg | 6.88 | 43.0 | 75.2 | 56.3 | 78.4 | 74.5 | 70.3 | 36.9 | 62.1 |
| | std | 0.01 | 0.7 | 0.5 | 0.3 | 0.3 | 1.0 | 0.6 | 1.1 | 0.3 |

*Table 15.* Zero-shot results of PuzzleMoE on Qwen1.5-MoE under 25% and 50% sparsity with different seeds.

| Sparsity | Seed | Wiki | ARC-c | ARC-e | Hella | Piqa | BoolQ | Wino | MMLU | Avg |
|---|---|---|---|---|---|---|---|---|---|---|
| 0% | - | 7.22 | 41.0 | 73.2 | 58.0 | 80.0 | 79.5 | 68.9 | 61.0 | 65.9 |
| 25% | 1 | 7.38 | 40.9 | 73.4 | 57.2 | 79.8 | 79.5 | 69.1 | 60.5 | 65.8 |
| | 2 | 7.37 | 40.7 | 74.0 | 57.2 | 80.2 | 80.1 | 70.9 | 60.1 | 66.2 |
| | 3 | 7.39 | 41.6 | 72.4 | 57.2 | 78.9 | 78.9 | 69.0 | 60.4 | 65.5 |
| | 4 | 7.37 | 40.8 | 73.8 | 57.2 | 79.4 | 79.4 | 69.4 | 60.5 | 65.8 |
| | 5 | 7.38 | 40.1 | 72.8 | 57.3 | 79.8 | 78.6 | 70.9 | 60.2 | 65.7 |
| | 6 | 7.38 | 41.2 | 74.8 | 57.4 | 79.7 | 79.4 | 68.7 | 60.5 | 66.0 |
| | 7 | 7.37 | 41.4 | 73.5 | 57.7 | 79.8 | 78.8 | 70.6 | 60.3 | 66.0 |
| | 8 | 7.38 | 40.9 | 74.0 | 57.4 | 79.8 | 78.6 | 68.4 | 60.3 | 65.6 |
| | 9 | 7.37 | 40.0 | 73.2 | 57.2 | 80.3 | 78.8 | 69.4 | 60.2 | 65.6 |
| | 10 | 7.37 | 40.8 | 73.2 | 57.4 | 79.7 | 79.2 | 69.1 | 60.4 | 65.7 |
| | 11 | 7.37 | 40.8 | 73.7 | 57.5 | 80.1 | 79.4 | 69.9 | 60.3 | 66.0 |
| | 12 | 7.37 | 40.7 | 73.3 | 57.3 | 79.4 | 79.4 | 70.2 | 60.9 | 65.9 |
| | 13 | 7.37 | 40.7 | 72.6 | 57.3 | 79.4 | 79.0 | 69.3 | 60.8 | 65.6 |
| | 14 | 7.35 | 41.8 | 73.6 | 57.3 | 79.7 | 78.8 | 69.6 | 60.8 | 65.9 |
| | 15 | 7.39 | 40.6 | 73.3 | 57.2 | 79.4 | 79.7 | 68.6 | 60.2 | 65.6 |
| | 16 | 7.38 | 41.0 | 73.2 | 57.2 | 79.2 | 78.8 | 69.9 | 60.4 | 65.7 |
| | avg | 7.37 | 40.9 | 73.4 | 57.3 | 79.7 | 79.2 | 69.6 | 60.4 | 65.8 |
| | std | 0.01 | 0.5 | 0.6 | 0.1 | 0.4 | 0.4 | 0.8 | 0.2 | 0.2 |
| 50% | 1 | 7.55 | 39.9 | 72.7 | 56.3 | 80.4 | 80.2 | 69.0 | 59.9 | 65.5 |
| | 2 | 7.54 | 40.2 | 74.0 | 56.5 | 79.3 | 79.9 | 68.3 | 59.7 | 65.4 |
| | 3 | 7.54 | 40.8 | 73.1 | 56.5 | 79.3 | 79.9 | 69.6 | 60.0 | 65.6 |
| | 4 | 7.54 | 40.5 | 74.0 | 56.4 | 78.7 | 78.9 | 68.7 | 60.0 | 65.3 |
| | 5 | 7.55 | 41.3 | 73.2 | 56.6 | 79.4 | 76.2 | 69.3 | 59.7 | 65.1 |
| | 6 | 7.54 | 40.7 | 74.3 | 56.3 | 79.3 | 79.6 | 69.1 | 59.8 | 65.6 |
| | 7 | 7.56 | 42.4 | 75.4 | 56.7 | 78.9 | 77.6 | 69.5 | 60.3 | 65.8 |
| | 8 | 7.54 | 40.6 | 74.1 | 56.5 | 79.6 | 78.4 | 69.8 | 60.0 | 65.6 |
| | 9 | 7.55 | 39.7 | 72.2 | 56.7 | 79.7 | 77.8 | 69.9 | 59.0 | 65.0 |
| | 10 | 7.56 | 40.7 | 72.9 | 56.6 | 79.5 | 78.1 | 70.1 | 60.5 | 65.5 |
| | 11 | 7.54 | 39.3 | 72.7 | 56.6 | 79.9 | 78.9 | 69.1 | 59.7 | 65.2 |
| | 12 | 7.54 | 39.9 | 73.2 | 56.5 | 78.9 | 78.0 | 69.7 | 60.5 | 65.2 |
| | 13 | 7.54 | 41.7 | 73.4 | 56.8 | 79.1 | 79.1 | 70.0 | 60.2 | 65.8 |
| | 14 | 7.54 | 41.6 | 74.2 | 56.7 | 79.3 | 77.3 | 68.4 | 60.0 | 65.4 |
| | 15 | 7.55 | 40.5 | 73.6 | 56.4 | 79.2 | 79.5 | 69.7 | 60.4 | 65.6 |
| | 16 | 7.55 | 40.9 | 73.4 | 56.6 | 79.4 | 78.7 | 70.4 | 59.5 | 65.6 |
| | avg | 7.55 | 40.7 | 73.5 | 56.5 | 79.4 | 78.6 | 69.4 | 60.0 | 65.4 |
| | std | 0.01 | 0.8 | 0.8 | 0.1 | 0.4 | 1.1 | 0.6 | 0.4 | 0.2 |

*Table 16.* Zero-shot results of PuzzleMoE on Qwen3-MoE under 25% and 50% sparsity with different seeds.

| Sparsity | Seed | Wiki | ARC-c | ARC-e | Hella | Piqa | BoolQ | Wino | MMLU | Avg |
|---|---|---|---|---|---|---|---|---|---|---|
| 0% | - | 8.70 | 52.7 | 79.3 | 59.5 | 79.6 | 88.7 | 70.4 | 77.8 | 72.6 |
| 25% | 1 | 9.17 | 49.3 | 76.5 | 57.9 | 79.2 | 87.6 | 71.0 | 76.6 | 71.2 |
| | 2 | 9.03 | 50.8 | 77.8 | 58.5 | 78.7 | 88.2 | 70.4 | 77.1 | 71.6 |
| | 3 | 9.07 | 51.8 | 79.3 | 58.4 | 79.2 | 87.9 | 68.8 | 76.1 | 71.6 |
| | 4 | 9.07 | 51.1 | 78.5 | 58.2 | 79.0 | 88.3 | 71.0 | 76.8 | 71.8 |
| | 5 | 9.12 | 54.3 | 79.2 | 58.5 | 79.8 | 87.8 | 70.2 | 76.4 | 72.3 |
| | 6 | 9.14 | 50.7 | 78.4 | 58.2 | 78.8 | 88.5 | 70.9 | 77.1 | 71.8 |
| | 7 | 9.17 | 51.9 | 79.3 | 58.4 | 79.3 | 88.3 | 69.5 | 76.3 | 71.9 |
| | 8 | 9.05 | 53.1 | 79.5 | 58.4 | 79.1 | 88.4 | 70.8 | 76.8 | 72.3 |
| | 9 | 9.09 | 51.0 | 78.9 | 58.1 | 79.1 | 88.3 | 70.4 | 76.5 | 71.8 |
| | 10 | 9.11 | 52.7 | 81.0 | 58.5 | 79.4 | 88.6 | 70.3 | 76.2 | 72.4 |
| | 11 | 9.04 | 51.5 | 78.8 | 58.2 | 79.3 | 87.7 | 69.9 | 76.9 | 71.8 |
| | 12 | 9.07 | 50.3 | 78.8 | 58.4 | 79.8 | 88.1 | 71.2 | 76.6 | 71.9 |
| | 13 | 8.97 | 51.5 | 79.3 | 58.1 | 80.0 | 88.8 | 69.8 | 76.5 | 72.0 |
| | 14 | 9.11 | 51.0 | 79.2 | 58.2 | 79.3 | 88.7 | 70.8 | 75.9 | 71.9 |
| | 15 | 9.11 | 50.7 | 78.2 | 58.2 | 78.6 | 88.0 | 70.9 | 76.8 | 71.6 |
| | 16 | 9.03 | 53.3 | 79.6 | 58.5 | 79.4 | 88.6 | 70.2 | 76.4 | 72.3 |
| | avg | 9.08 | 51.6 | 78.9 | 58.3 | 79.3 | 88.2 | 70.4 | 76.6 | 71.9 |
| | std | 0.05 | 1.3 | 1.0 | 0.2 | 0.4 | 0.4 | 0.6 | 0.3 | 0.3 |
| 50% | 1 | 9.46 | 50.1 | 77.7 | 56.9 | 78.7 | 87.0 | 70.4 | 74.7 | 70.8 |
| | 2 | 9.57 | 54.8 | 79.4 | 57.0 | 78.7 | 87.8 | 69.5 | 75.3 | 71.8 |
| | 3 | 9.54 | 50.9 | 77.8 | 56.8 | 78.9 | 87.4 | 69.7 | 75.1 | 70.9 |
| | 4 | 9.46 | 49.5 | 77.0 | 57.1 | 79.1 | 87.9 | 71.4 | 75.5 | 71.1 |
| | 5 | 9.59 | 52.3 | 79.1 | 56.8 | 79.1 | 88.0 | 70.8 | 75.5 | 71.7 |
| | 6 | 9.53 | 50.6 | 78.0 | 57.3 | 78.7 | 88.0 | 70.2 | 74.9 | 71.1 |
| | 7 | 9.45 | 50.2 | 79.4 | 57.3 | 78.2 | 88.1 | 70.6 | 74.9 | 71.2 |
| | 8 | 9.51 | 50.4 | 78.3 | 56.9 | 79.1 | 88.4 | 70.6 | 75.2 | 71.3 |
| | 9 | 9.53 | 51.6 | 78.0 | 57.3 | 79.0 | 88.1 | 70.0 | 74.7 | 71.2 |
| | 10 | 9.47 | 49.5 | 79.6 | 57.0 | 78.9 | 88.0 | 68.8 | 74.4 | 70.9 |
| | 11 | 9.51 | 50.9 | 79.0 | 57.1 | 78.9 | 87.9 | 69.8 | 75.2 | 71.3 |
| | 12 | 9.44 | 50.7 | 79.6 | 57.3 | 78.5 | 87.9 | 70.6 | 75.6 | 71.5 |
| | 13 | 9.46 | 50.5 | 78.5 | 56.9 | 78.8 | 88.1 | 69.9 | 74.9 | 71.1 |
| | 14 | 9.43 | 52.4 | 79.8 | 57.4 | 79.9 | 88.4 | 69.5 | 75.6 | 71.9 |
| | 15 | 9.58 | 47.4 | 76.6 | 57.3 | 78.4 | 87.7 | 68.9 | 74.7 | 70.1 |
| | 16 | 9.53 | 53.9 | 78.5 | 57.1 | 79.2 | 88.8 | 70.1 | 75.2 | 71.8 |
| | avg | 9.50 | 51.0 | 78.5 | 57.1 | 78.9 | 88.0 | 70.1 | 75.1 | 71.2 |
| | std | 0.05 | 1.8 | 1.0 | 0.2 | 0.4 | 0.4 | 0.7 | 0.4 | 0.4 |

