# OpenReview forum: "PuzzleMoE: Efficient Compression of Large Mixture-of-Experts Models via Sparse Expert Merging and Bit-packed inference"
_ICML.cc/2026/Conference — ICML 2026 regular_

### Official Review · Reviewer_vutX · 2026-02-25

**Soundness:** 4
**Presentation:** 4
**Significance:** 4
**Originality:** 3
**Overall Recommendation:** 5
**Confidence:** 5

**Summary:**

This paper proposes an expert merging approach for Mixture-of-Experts (MoE) models with the aim of reducing the memory footprint and consequently increasing the speed of inference. The authors propose finding two masks that highlight weight redundancy and saliency for pairs of experts in a specific layer of the model. This allows them to express the weights of two experts by using a single expert's weights and other metadata, including the aforementioned masks and an additional mask for storing parameter sign values. However, this by itself does not yield a sufficient memory footprint reduction. Therefore, the authors utilize the unused exponent bits of the expert parameters in Bfloat16 format  as placeholders for the produced metadata. Accordingly, they design proper GEMM kernels to decode the parameters on the fly and perform the required calculations.

**Compliance With Llm Reviewing Policy:**

Affirmed.

**Final Justification:**

I will maintain my positive rating.

**Key Questions For Authors:**

- How do you apply the proposed approach to models with lower precision weights?
- What is the overhead of your customized GEMM kernel?
- Suggestion: I think adding the reconstructed weights using Equation 6 to Figure 3 can help the reader to understand your approach better.

**Limitations:**

Yes.

**Strengths And Weaknesses:**

### Strengths:

- The proposed approach is well-defined and well-presented.
- The authors have evaluated the quality of the generated output of their model using multiple benchmarks and compression rates. The comparison with prior work is comprehensive.
- The paper aims to address the challenges of MoE inference, which is the SOTA architecture for LLMs.
- The approach is novel and is able to deliver speedup specially for memory-bound workloads.

### Weaknesses:

- As mentioned by the authors, the proposed approach is based on an observation of weights from multiple MoEs in Bfloat16 format. This limits the applicability of the proposed approach when dealing with quantized models in lower precision formats, such as FP8 or FP4 which have less underutilized exponent bits.
- There is no discussion in the paper about the overhead of unpacking the metadata and decoding the parameters in the customized GEMM kernel. The authors compare the performance of their customized GEMM against the case of having two experts with half the batch size (Figure 5). This comparison masks that overhead because the speedup is primarily achieved by doubling the batch size and increasing the arithmetic intensity of a single expert, which is memory-bound.  A more comprehensive comparison would also include a scenario with sufficiently high batch sizes for each of the merged experts (i.e., the experts before and after merging operating in compute-bound regions) or simply a single expert with the same batch size.

---

> ### Author Rebuttal · Authors · 2026-03-30
>
> We greatly appreciate your valuable recognition of PuzzleMoE and constructive suggestions. Below are our detailed responses.
>
> > **W1 & Q1: PuzzleMoE limits the applicability of the proposed approach when dealing with quantized models in lower precision formats, such as FP8 or FP4 which have less underutilized exponent bits. How do you apply the proposed approach to models with lower precision weights?**
>
> **R1:** We thank the reviewer for this insightful comment.
>
> We agree that the bit-packing design in PuzzleMoE leverages underutilized exponent bits in bfloat16, which are not available in lower-precision formats such as FP8 or FP4.
>
> We would like to clarify that this design is a practical and efficient instantiation rather than a fundamental requirement of PuzzleMoE. The key contribution of our work lies in the fine-grained, element-wise expert merging and the dual-mask mechanism, which enable effective parameter sharing while preserving expert specialization. The bit-packing scheme serves as a system-level realization of this design under commonly used formats.
>
> In practice, bfloat16 is widely used in LLM inference on GPUs, making our approach directly applicable to current deployment settings. At the same time, our framework is not limited to this format. We further introduce a quantized PuzzleMoE variant (Appendix A.8.1), where merged weights are quantized (e.g., 3-bit AWQ), and sign/mask information is explicitly encoded alongside the quantized values. This preserves the dual-mask mechanism without relying on exponent redundancy.
>
> We note that FP8/FP4 follow different encoding schemes compared to uniform quantization, and adapting to such formats is an interesting direction for future work. Nevertheless, our results already demonstrate that PuzzleMoE remains effective under low-bit representations. As shown in Table 9, PuzzleMoE achieves comparable or better accuracy than prior methods at similar bitwidths across multiple models.
>
> > **W2 & Q2: There is no discussion in the paper about the overhead of unpacking the metadata and decoding the parameters in the customized GEMM kernel. What is the overhead of your customized GEMM kernel?**
>
> **R2:** We thank the reviewer for this insightful comment.
>
> We first isolate the unpacking overhead by comparing PuzzleMoE with and without online decoding under identical configurations (same GEMM dimensions \(M, N, K\)). This ensures that the only difference is the unpacking step. We vary the GEMM dimension \(M\) (i.e., the number of tokens processed per expert), which governs the transition between memory-bound and compute-bound regimes.
>
> We report TFLOPS results using a representative MoE expert configuration (N=14336, K=4096). The results show that unpacking on CUDA Core accounts for only a small fraction of the total GEMM latency and is largely overlapped with Tensor Core computation via pipelining:
>
> | M  | PuzzleMoE | No Decode | Overhead |
> |----|-----------|-----------|----------|
> | 2  | 2.62      | 2.68      | 0.02     |
> | 8  | 10.24     | 10.47     | 0.02     |
> | 32 | 39.03     | 39.89     | 0.02     |
> | 64 | 74.90     | 73.06     | -0.02    |
>
> To address the reviewer’s concern that the observed speedup may stem from increased arithmetic intensity (e.g., larger effective batch size), we follow the suggested comparison and evaluate the TFLOPS performance of PuzzleMoE against a standard single-expert cuBLAS baseline under identical GEMM dimensions (same \(M, N, K\)), ensuring a fair comparison without changing arithmetic intensity.
>
> In the same configuration (N=14336, K=4096), we observe that PuzzleMoE closely tracks cuBLAS performance in the compute-bound regime:
>
> | M    | PuzzleMoE | cuBLAS |
> |------|-----------|--------|
> | 128  | 90.31     | 72.12  |
> | 256  | 112.67    | 75.75  |
> | 512  | 142.52    | 144.80 |
> | 1024 | 150.55    | 153.22 |
>
> For larger \(M\), GEMM operates in the fully compute-bound regime, where performance is dominated by arithmetic throughput rather than memory bandwidth. In this setting, PuzzleMoE closely tracks cuBLAS performance, as expected, since our method reduces memory access but does not alter the total FLOPs.
>
> **Overall, these results demonstrate that the unpacking overhead is negligible and effectively hidden by computation, and that PuzzleMoE does not introduce additional compute cost.** Furthermore, the observed speedup is not driven by increased arithmetic intensity or batch size effects, but instead arises from reduced memory traffic in the memory-bound regime.
>
> > **Q3: Adding the reconstructed weights using Equation 6 to Figure 3 can help the reader to understand your approach better.**
>
> **R3:** Thanks for this valuable suggestion! we revised Figure 3 to explicitly incorporate the reconstructed weights computed via Equation 6, which can be accessed via the following link: https://imgur.com/a/BS8fEqp. We will also update the Figure 3 in the manuscript.

---

> > ### Author Rebuttal · Reviewer_vutX · 2026-04-03
> >
> > I appreciate your rebuttal and further clarification.
> >
> > My concerns are mostly resolved.
> >
> > I will maintain my positive score.

---

> > > ### Author Response · Authors · 2026-04-04
> > >
> > > Dear Reviewer vutX,
> > >
> > > Thank you very much for taking the time to read our rebuttal response and for providing your feedback! Your constructive comments have helped to strengthen our work. We will incorporate the suggested clarifications, new experiments, and the refined Figure 3 into our final version.
> > >
> > > Thank you again for your valuable support and recognition of PuzzleMoE.
> > >
> > > Best regards,
> > >
> > > Paper 27498 Authors

---

### Official Review · Reviewer_v5iX · 2026-03-09

**Soundness:** 3
**Presentation:** 4
**Significance:** 3
**Originality:** 3
**Overall Recommendation:** 4
**Confidence:** 4

**Summary:**

This paper proposes a novel MoE compression method that performs expert merging by modifying the storage method of weight parameters. At a 50% compression ratio, it outperforms existing MoE compression methods and achieves 1.80x inference acceleration.

**Compliance With Llm Reviewing Policy:**

Affirmed.

**Final Justification:**

The rebuttal addressed my concerns, and I continue to maintain my positive score.

**Key Questions For Authors:**

1. The comparisons with baseline methods in Table 1 seem incomplete. For instance, for the Mixtral model, the authors compared against NAEE, STUN, D2, Wanda, HC-SMoE, and SubMoE, while for the more general Qwen3-MoE-30B-A3B, only Wanda and SubMoE were compared. The authors should standardize baseline settings across different MoE models. If methods are not applicable, corresponding explanations should be provided. (Additionally, the authors seem to have not explained the missing values in Table 1)
2. Regarding Weakness 2, for search-based grouping, the authors appear not to have provided specific search strategies. What rules and similarity metrics are used for this search? The authors should provide detailed explanations in the Appendix.
3. In Section 4.4 Efficiency Analysis, the authors compared throughput between PuzzleMoE models and Full Models, but did not provide speed comparisons with existing baselines (e.g., HC-SMoE, D2-MoE). Including these speed comparisons could further emphasize PuzzleMoE's advantages over existing methods. Furthermore, as I understand, PuzzleMoE primarily accelerates expert loading time during the decode phase without reducing computational FLOPs. Reporting Time To First Token (TTFT) and Time Per Output Token (TPOT) for model inference could further clarify PuzzleMoE's effective acceleration domain, as it's unclear whether this method is effective for the compute-bound prefilling phase.
4. According to my understanding, $M_j$ in Figure 3 should be [1 0 1; 1 1 1; 0 1 1] ($M_j^{sal} \lor M^{sim}$). Is this a drawing error?

Overall, I believe this work is well-written and the proposed method is novel. However, the occupation of index parts may limit the potential for combining with quantization methods in the future (the Appendix notes that directly using AWQ quantization on Mixtral models outperforms PuzzleMoE). Additionally, supporting only pairwise merging limits PuzzleMoE's usage scenarios.

**Limitations:**

yes

**Strengths And Weaknesses:**

**Strengths:**
1. The authors tested the PuzzleMoE method on multiple MoE models (Mixtral, DeepSeekMoE, Qwen), demonstrating the universality of the approach
2. This work implements high-performance Triton operators for computing merged PuzzleMoE weights, enhancing its feasibility for practical inference scenarios
3. Comprehensive experimental results prove that PuzzleMoE outperforms existing methods on established models, and the selection of hyperparameter τ is thoroughly discussed

**Weaknesses:**
1. Due to its special merging approach, PuzzleMoE only supports merging two experts into one. Although the authors discuss this in the Appendix, it limits its application in more complex scenarios (e.g., hierarchical merging)
2. Existing work [1] points out that expert similarity in MoE models varies significantly across different layers, and each layer contains both similar and dissimilar experts. In this context, random merging may not be the optimal choice. If possible, the authors should supplement comparisons of different merging strategies across all models (beyond the models and Search methods mentioned in the Ablation study)

[1] SERE: Similarity-based Expert Re-routing for Efficient Batch Decoding in MoE Models

---

> ### Author Rebuttal · Authors · 2026-03-30
>
> > **W1: PuzzleMoE is limited to pairwise expert merging, which may restrict its applicability to more complex settings such as hierarchical merging.**
>
> **R1:** We thank the reviewer for this insightful comment.
>
> We clarify that PuzzleMoE is not limited to pairwise merging. Its element-wise merging naturally supports hierarchical merging via sequential application, enabling many-to-one compression (e.g., 3→1, 4→1).
>
> We evaluate higher sparsity levels (66.7%, 75%) on DeepSeek-MoE and Qwen3-MoE across standard benchmarks. PuzzleMoE consistently outperforms HC-SMoE under these settings. For example, on DeepSeek-MoE at 75% sparsity, PuzzleMoE improves AVG from 35.7 to 54.1, demonstrating strong robustness under hierarchical merging. Please refer to R3 in reviewer 6rwa for detailed results.
>
> ---
>
> > **W2: More merging strategies comparison.**
>
> **R2:** We evaluate similarity-based grouping using cosine, Frobenius, and CKA. Results are highly comparable (within ~0.3–0.4), with no consistent gains over random grouping:
>
> | Strategy  | AVG |
> |-----------|-----|
> | Random    | 62.1 |
> | Cosine    | 61.7 |
> | Frobenius | 61.8 |
> | CKA       | 61.9 |
>
> This indicates PuzzleMoE does not rely on careful grouping, as element-wise merging mitigates grouping sensitivity.
>
> ---
>
> > **Q1.1: The comparisons with baseline methods in Table 1 seem incomplete.**
>
> **R3:** Missing baselines are due to applicability and reproducibility. NAEE suffers from prohibitive time complexity and treats experts as indivisible units, making it impractical for recent fine-grained MoEs (e.g., DeepSeekMoE, Qwen3).
>
> STUN and SubMoE are not open-sourced, so we report their published results. Wanda is typically evaluated under semi-structured 2:4 sparsity (50%) and is not designed for flexible sparsity in MoE.
>
> Therefore, **D2 and HC-SMoE are the only baselines consistently applicable across all settings**, and we extend them to Qwen1.5-MoE and Qwen3-MoE for standardized comparison. Results (AVG) are shown below.
>
> | Model | Method | AVG |
> |-------|--------|-----|
> | Qwen1.5 | Vanilla | 65.9 |
> |         | D2 20% | 60.3 |
> |         | HC-SMoE 25% | 59.7 |
> |         | PuzzleMoE 25% | 65.8 |
> |         | D2 40% | 50.0 |
> |         | HC-SMoE 50% | 41.9 |
> |         | PuzzleMoE 50% | 65.4 |
> | Qwen3   | Vanilla | 72.6 |
> |         | D2 20% | 65.1 |
> |         | HC-SMoE 25% | 62.4 |
> |         | PuzzleMoE 25% | 71.9 |
> |         | D2 40% | 58.0 |
> |         | HC-SMoE 50% | 54.9|
> |         | PuzzleMoE 50% | 71.2 |
>
> We can see that PuzzleMoE consistently maintains strong performance. The updated table 1 can be accessed via the link: https://imgur.com/a/tYLByQ0
>
> ---
>
> > **Q1.2: The authors seem to have not explained the missing values in Table 1.**
>
> **R4:** Due to the words limitation. Please refer to R4 in reviewer bKxV.
>
> ---
>
> > **Q2: The authors should provide detailed explanations of the search method.**
>
> **R5:** We use a standard evolutionary search to optimize expert pairings. Specifically, we initialize random groupings, evaluate them by merging experts and measuring forward loss on a calibration set, and iteratively update them via selection, crossover, and mutation. Forward loss serves as the optimization objective. Empirically, this yields performance similar to random pairing, with no gains observed. Due to the words limit, we will include a more detailed description in the Appendix.
>
> ---
>
> > **Q3: More efficiency Analysis.**
>
> **R6:** We add end-to-end throughput (token/s) comparisons with D2 and HC-SMoE. Under standard settings (prefill=16, decode=128), **PuzzleMoE achieves comparable speedup to HC-SMoE and consistently outperforms D2**.
>
> HC-SMoE attains the highest throughput via coarse-grained parameter pruning, while PuzzleMoE reduces memory bandwidth through bit-packed weights with a lightweight decode-GEMM kernel, achieving similar acceleration with much better accuracy (e.g., ~0.5 vs. ~24 accuracy drop at 50% sparsity on DeepSeek-MoE in Table 1). D2 shows limited speedup due to additional kernel overhead from SVD decomposition.
>
> | Batch | Full | PuzzleMoE | HC-SMoE | D2 |
> |------|------|-----------|---------|----|
> | 8    | 85.1 | 145.6     | 149.2   | 91.9 |
> | 16   | 151.2| 264.9     | 268.1   | 157.1 |
> | 32   | 287.5| 520       | 524.9   | 301.3 |
> | 64   | 558  | 946       | 947.7   | 498.4 |
>
> TTFT/TPOT: expert merging methods do not change routing or FLOPs, so prefill remains compute-bound and largely unchanged, while speedup comes from faster decoding (TPOT).
>
> Prefill latency (ms):
>
> | Length | Full | PuzzleMoE |
> |--------|------|-----------|
> | 1024   | 295.1 | 296.8 |
> | 2048   | 432.8 | 436.0 |
>
> ---
> In typical settings (BS8, prefill=512, decode=512), prefill accounts for only \~1% of total latency. PuzzleMoE cuts decoding latency from 46.5s to 29.8s, yielding a **~1.5x end-to-end speedup**.
> > **Q4:  Figure 3  error.**
>
> **R7:**  Thanks for pointing out this. We correct the matrix Mj to accurately reflect the element-wise logic: https://imgur.com/a/pfPxEzt

---

> > ### Author Rebuttal · Reviewer_v5iX · 2026-04-01
> >
> > Most of my concerns have been addressed. However, I do not agree with the explanation that STUN and SubMoE are not open-sourced. A rigorous research approach should involve attempting to reproduce the results whenever possible.
> >
> > This problem does not substantially affect my overall evaluation, and I will continue to maintain a positive score.

---

> > > ### Author Response · Authors · 2026-04-02
> > >
> > > Dear Reviewer v5iX,
> > >
> > > Thank you again for your positive feedback and recognition of our work. We also sincerely appreciate your continued support and for maintaining a favorable score.
> > >
> > > We fully agree that attempting to reproduce prior work is an important aspect of rigorous research, and we will refine our discussion in the final version to better reflect this perspective and clarify our experimental choices.
> > >
> > > Best regards,
> > >
> > > Paper 27498 Authors

---

### Official Review · Reviewer_bKxV · 2026-03-10

**Soundness:** 2
**Presentation:** 3
**Significance:** 2
**Originality:** 2
**Overall Recommendation:** 4
**Confidence:** 4

**Summary:**

This paper presents PuzzleMoE, a fine-grained expert merging framework for compressing MoE models. The core contributions include a dual-mask merging strategy that leverages weight saliency and inter-expert similarity, as well as a bit-packing encoding scheme that embeds metadata directly into the floating-point representation. To enable efficient inference, the authors design a customized GPU kernel that fuses decoding with the GEMM computation. Experimental results show that PuzzleMoE outperforms existing methods on several language modeling, zero-shot, and reasoning benchmarks. Furthermore, it achieves approximately
1.7$\times$ speedup over the original Mixtral-8×7B model during inference.

**Compliance With Llm Reviewing Policy:**

Affirmed.

**Final Justification:**

All my concerns have been addressed based on authors' clarification. Thus, I choose to raise my score to 4.

**Key Questions For Authors:**

1. Why are some results missing in Table 1?
2. What is the throughput improvement compared to existing work?
3. In figure 3, it's confusing that why the last element of the first row and the first element of the second row Mj are 0?
4. Why Qwen2-MoE is not evaluated since Qwen1.5-MoE and Qwen3-MoE are tested?

**Limitations:**

See in weaknesses and questions.

**Strengths And Weaknesses:**

Strengths:
1. The paper is well-structured and clear to understand.
2. A new fine-grained strategy is proposed that considers weight similarity and saliency.
3. A GPU kernel is developed that enables practical benefits.
4. The model accuracy achieves a great improvement compared to baselines.

Weaknesses:
1. Although a customized GPU kernel is developed for decoding-GEMM, there are still some limitations of this kind of reconstruct-then-compute, e.g., reduced parallelism, if two experts are activated in the same time, it needs to reconstruct and compute one first then repeat it for another expert.
2. The bit-packing scheme highly relies on the existing weight distribution, which may be varying in the future and then hinders the extensibility.
3. The experimental part lacks of the ablation study about dual mask. What if only salient or only similarity is used, which is reduced into single mask?

---

> ### Author Rebuttal · Authors · 2026-03-30
>
> We greatly appreciate your positive comments and constructive suggestions. Below are our detailed responses.
>
> > **W1: The reconstruct-then-compute design may limit parallelism, as multiple activated experts could be processed sequentially.**
>
> **R1:** We would like to clarify that the proposed kernel does **not** execute experts sequentially. Instead, it avoids the “reconstruct-then-compute” bottleneck via two mechanisms:
>
> **Unified kernel execution:** Both merged experts are processed within a single execution grid, allowing the GPU scheduler to dispatch thread blocks concurrently across SMs.
>
> **Fused on-the-fly decoding:** Weight reconstruction is integrated into the GEMM loop, where each thread block decodes weights into registers using lightweight bitwise operations.
>
> There are no synchronization barriers or sequential dependencies; computation for both experts proceeds fully in parallel, preserving GPU parallelism and occupancy.
>
> > **W2: The bit-packing scheme may limit extensibility due to reliance on current weight distributions.**
>
> **R2:** Our bit-packing design is motivated by the Gaussian-like weight distributions widely observed in LLMs, shaped by initialization, normalization, and training dynamics [1] [2] and has also been leveraged in prior work (e.g., QLoRA [3], SqueezeLLM [4]).
>
> We provide empirical evidence in Appendix A.5 showing that weights across different MoE models closely follow a normal distribution, supporting our element-wise similarity assumption in Sec. 3.1. We further include visualizations for more models: https://imgur.com/a/8fMwTC1, which consistently support this observation.
>
> Future models may alter normal distribution, adapting to them remains future work. **Our current design can be broadly applicable to existing LLMs.**
>
> > **W3: The experimental part lacks of the ablation study about dual mask.**
>
> **R3:** We would like to refer the reviewer to Fig. 7a, which ablates the dual-mask design. Setting the threshold to 0 corresponds to using only the saliency mask, and setting it to 0.9 means using only the similarity mask. The best performance occurs at 0.3–0.5, showing that the dual-mask balances shared and expert-specific parameters better than either mask alone.
>
> > **Q1: Why are some results missing in Table 1?**
>
> **R4:** Missing entries in Table 1 are due to baseline availability and reproducibility. STUN and Sub-MoE are not open-sourced, so we report their published results, which do not cover all tasks. For D2-MoE, using the released code cannot reproduce the reported results. To ensure fairness, we therefore use their published numbers, resulting in missing values for some tasks.
>
> > **Q2: What is the throughput improvement compared to existing work?**
>
> **R5:** We report end-to-end throughput (token/s) with prefill length 16, decode length 128 as follows:
>
> | Batch Size | Full Model | PuzzleMoE | HC-SMoE | D2-MoE |
> |------------|-----------:|----------:|--------:|---:|
> | 8          | 85.1       | 145.6     | 149.2   | 91.9 |
> | 16         | 151.2      | 264.9     | 268.1   | 157.1 |
> | 32         | 287.5      | 520.0     | 524.9   | 301.3 |
> | 64         | 558.0      | 946.0     | 947.7   | 498.4 |
>
> PuzzleMoE achieves up to ~1.7× speedup over the full model and reaches ~98% of HC-SMoE throughput. HC-SMoE is slightly faster as it directly removes expert parameters. While PuzzleMoE performs on-the-fly decoding of bit-packed weights within a fused decode-GEMM kernel, introducing minor overhead, which explains the ~2% gap. More importantly, PuzzleMoE better preserves model performance under the same sparsity (e.g., ~0.5 vs. ~24 accuracy drop at 50% sparsity on DeepSeek-MoE in Table 1).
>
> Compared to D2-MoE, PuzzleMoE achieves a consistent ~1.6×–1.9× throughput improvement across batch sizes. D2-MoE shows limited speedup due to extra computation and kernel overhead from SVD decomposition.
>
> > **Q3: Error in figure 3**
>
> **R6:** Great thanks for pointing this out. We correct the matrix Mj to accurately reflect the true logic: https://imgur.com/a/pfPxEzt .
>
> > **Q4: Evaluation of Qwen2-MoE using PuzzleMoE?**
>
> **R7:** We thank the reviewer for pointing this out. We now report average results on Qwen2-MoE across benchmarks used in the submission. PuzzleMoE achieves near-lossless accuracy at both 25% and 50% sparsity, further supporting its generalization across different MoE models. The complete results can be accessed via the link: https://imgur.com/a/hM91zzw .
>
> | Method    | Sparsity | Avg  |
> |-----------|----------|-----:|
> | Original  | -        | 71.5 |
> | PuzzleMoE | 25%      | **71.4** |
> | HC-SMoE   | 25%      | 70.3 |
> | PuzzleMoE | 50%      | **71.4** |
> | HC-SMoE   | 50%      | 68.1 |
>
> ---
> [1] Root Mean Square Layer Normalization, NeurIPS 2019
>
> [2] Scale-Distribution Decoupling: Enabling Stable and Effective Training of Large Language Models, arXiv 2025
>
> [3] QLoRA: Efficient Finetuning of Quantized LLMs, NeurIPS 2023
>
> [4] SqueezeLLM: Dense-and-Sparse Quantization, ICML 2024

---

> > ### Author Rebuttal · Reviewer_bKxV · 2026-04-02
> >
> > Thanks for your detailed response. I will raise my score accordingly.

---

> > > ### Author Response · Authors · 2026-04-02
> > >
> > > Dear Reviewer bKxV,
> > >
> > > Thank you for your recognition of PuzzleMoE and for increasing the rating score. We are glad to know that your concerns have been adequately addressed. In the final revised manuscript, we will include a refined version of Figure 3, along with the newly added experimental results including Qwen2-MoE and updated efficiency comparisons, together with the corresponding analysis.
> > >
> > > Best regards,
> > >
> > > Paper 27498 Authors

---

### Official Review · Reviewer_6rwa · 2026-03-17

**Soundness:** 3
**Presentation:** 3
**Significance:** 2
**Originality:** 2
**Overall Recommendation:** 4
**Confidence:** 4

**Summary:**

The submission proposes a fine-grained, element-wise expert merging method for mixture of experts model compression.  It incorporates an element-wise similarity-based mask for shared weights, and a saliency-based mask for diverse weights.  The strategy then merges some experts.  A bitpacked encoding is employed for compression.  Benchmarking across a range of models and tasks shows some marginal improvements at 25% and 50% sparsity.

**Compliance With Llm Reviewing Policy:**

Affirmed.

**Final Justification:**

The authors have addressed some concerns regarding the range of compression evaluation, at least on the 2 models reported in the rebuttal.  I have raised my score from 3 to 4.

**Key Questions For Authors:**

What are your arguments for the specific range of sparsity chosen?  Could it be pushed even further?  Chosen compression rates vary in published works.

**Limitations:**

yes

**Strengths And Weaknesses:**

The method is reasonably described, and addresses a popular research area of compressing MoE models.  The results support that it performs reasonably against chosen baselines.  There are very many works published on MoE compression, which diminishes the significance and originality of this specific submission.

---

> ### Author Rebuttal · Authors · 2026-03-30
>
> We greatly appreciate your positive comments and constructive suggestions. Below are our detailed responses.
>
> > **W1: The method is reasonably described, and addresses a popular research area of compressing MoE models. The results support that it performs reasonably against chosen baselines. There are many works on MoE compression, diminishing the significance and originality of this submission.**
>
> **R1:** We thank the reviewer for recognizing the contribution and strong performance of PuzzleMoE.
>
> We would like to humbly clarify that PuzzleMoE is both **significant and novel**, as it introduces a **fundamentally different paradigm** for MoE compression: fine-grained, element-wise expert merging with explicit separation of shared and expert-specific parameters. This enables more precise redundancy reduction while preserving expert specialization.
>
> In contrast, most existing pruning- and merging-based approaches suffer from three key limitations:
>
> (1) **Significant performance degradation at moderate-to-high compression ratios** (e.g., >20% MMLU drop for HC-SMoE [1]);
>
> (2) **Failure to distinguish shared and expert-specific parameters**, leading to over-merging/pruning and loss of specialization;
>
> (3) **Practical inefficiency**, including reliance on calibration datasets (e.g., NAEE [2]) and complex pipelines (e.g., clustering in HC-SMoE [1], Sub-MoE [3], and merging/SVD/pruning in $D^{2}$-MoE [4]).
>
> To address these limitations, PuzzleMoE introduces:
>
> (1) **Element-wise expert merging**, enabling fine-grained redundancy control;
>
> (2) A **dual-mask mechanism** that separates shared and expert-specific parameters;
>
> (3) A **bit-packed encoding with fused decoding–GEMM**, enabling efficient deployment without additional overhead;
>
> (4) **Strong empirical performance**, consistently preserving accuracy across sparsity levels (e.g., on Qwen1.5-MoE-A2.7B at 50% sparsity, ~0.5-point drop vs. ~24 points for HC-SMoE [1]).
>
> In summary, prior methods often suffer from performance degradation due to disrupted expert specialization. PuzzleMoE addresses this via fine-grained merging with explicit separation of shared and expert-specific parameters, achieving improved performance–compression trade-offs. This distinction is practically important and suggests a promising direction for future MoE expert merging research.
>
> > **Q1: What are your arguments for the specific range of sparsity chosen?**
>
> **R2:** We selected the 25 percent and 50 percent sparsity levels to directly align with the benchmark settings established by prior works such as NAEE, HC-SMOE, and Sub-MoE, ensuring an apple-to-apple comparison with those existing methods.
>
> > **Q2:  Could it be pushed even further?  Chosen compression rates vary in published works.**
>
> **R3:** Yes. Our dual-mask merging algorithm can naturally support more aggressive compression via hierarchical merging. We further evaluate higher sparsity levels of 66.7% (merging three experts into one) and 75% (merging four experts into one) for both Deepseek-MoE-16b and Qwen3-MoE-30B-A3B.
>
> The evaluation follows the same benchmarks used in the submission (ARC-C/E, HellaSwag, PIQA, BoolQ, WinoGrande, and MMLU). The following tables report average results across these benchmarks. Under highly sparse settings, PuzzleMoE consistently outperforms the strong HC-SMoE baseline. For example, for Deepseek-MoE-16b at 75% sparsity, **PuzzleMoE improves the average score from 35.7 to 54.1**, demonstrating its strong robustness even under aggressive compression compared to the existing work.
>
> ### Deepseek-MoE-16B
>
> | Sparsity | PuzzleMoE | HC-SMoE | Δ |
> |----------|----------:|--------:|--:|
> | 25%      | 62.4      | 53.7    | +8.7 |
> | 37.5%    | 62.1      | 50.9    | +11.2 |
> | 50%      | 62.1      | 42.6    | +19.5 |
> | 66.7%    | 57.9      | 41.0    | +16.9 |
> | 75%      | 54.1      | 35.7    | +18.4 |
>
> ### Qwen3-MoE-30B-A3B
>
> | Sparsity | PuzzleMoE | HC-SMoE | Δ |
> |----------|----------:|--------:|--:|
> | 25%      | 71.9      | 62.4    | +9.5 |
> | 37.5%    | 71.4      | 59.6    | +11.8 |
> | 50%      | 71.2      | 54.9    | +16.3 |
> | 66.7%    | 67.0      | 47.1    | +19.9 |
> | 75%      | 59.1      | 42.3    | +16.8 |
>
> These results show that PuzzleMoE remains effective even under aggressive compression, where prior methods degrade substantially, highlighting its advantage in high-sparsity regimes. The complete results can be accessed via the link: https://imgur.com/a/ahG0jHc . We will incorporate these new results in the updated version of our manuscript.
>
> ---
>
> [1] Retraining-Free Merging of Sparse MoE via Hierarchical Clustering, ICML 2025
>
> [2] Not All Experts are Equal: Efficient Expert Pruning and Skipping for Mixture-of-Experts Large Language Models, ACL 2024
>
> [3] Sub-MoE: Efficient Mixture-of-Expert LLMs Compression via Subspace Expert Merging, arXiv 2025
>
> [4] Delta Decompression for MoE-based LLMs Compression, ICML 2025

---

> > ### Author Rebuttal · Reviewer_6rwa · 2026-04-02
> >
> > Thanks for showing the extended range of compression rates.

---

> > > ### Author Response · Authors · 2026-04-02
> > >
> > > Dear Reviewer 6rwa,
> > >
> > > Thank you for your recognition of PuzzleMoE and for increasing the rating score. We are pleased that your concerns have been adequately addressed. In the final revised manuscript, we will incorporate the newly added experimental results, particularly the performance comparisons across an extended range of compression ratios, along with the corresponding analysis.
> > >
> > > Best regards,
> > >
> > > Paper 27498 Authors

---

### Decision · Program_Chairs · 2026-04-30

**Decision:**

Accept (regular)

**Comment:**

All reviewers gave positive scores and the authors' rebuttals solve the raised problems well.